# GTR: A General, Multi-View, and Dynamic Framework for Trajectory Representation Learning

Xiangheng Wang [1]   Ziquan Fang [1]   Chenglong Huang [1]   Danlei Hu [2]   Lu Chen [2]   Yunjun Gao [2]

## Abstract

Trajectory representation learning aims to transform raw trajectory data into compact and low-dimensional vectors that are suitable for downstream analysis. However, most existing methods adopt either a free-space view or a road-network view during the learning process, which limits their ability to capture the complex, multi-view spatiotemporal features inherent in trajectory data. Moreover, these approaches rely on task-specific model training, restricting their generalizability and effectiveness for diverse analysis tasks. To this end, we propose GTR, a general, multi-view, and dynamic Trajectory Representation framework built on a pre-train and fine-tune architecture. Specifically, GTR introduces a multi-view encoder that captures the intrinsic multi-view spatiotemporal features. Based on the pre-train and fine-tune architecture, we provide the spatio-temporal fusion pre-training with a spatio-temporal mixture of experts to dynamically combine spatial and temporal features, enabling seamless adaptation to diverse trajectory analysis tasks. Furthermore, we propose an online frozen-hot updating strategy to efficiently update the representation model, accommodating the dynamic nature of trajectory data. Extensive experiments on two real-world datasets demonstrate that GTR consistently outperforms 15 state-of-the-art methods across 6 mainstream trajectory analysis tasks. All source code and data are available at https://github.com/ZJU-DAILY/GTR.

---
*Equal contribution  [1]School of Software, Zhejiang University, Ningbo, China  [2]College of Computer Science, Zhejiang University, Hangzhou, China. Correspondence to: Ziquan Fang <zq-fang@zju.edu.cn>.

*Proceedings of the 42nd International Conference on Machine Learning*, Vancouver, Canada. PMLR 267, 2025. Copyright 2025 by the author(s).

## 1. Introduction

With the widespread use of GPS devices and location-based services, large volumes of trajectory data have been collected (Wang et al., 2020; Zhou et al., 2024). A trajectory is typically represented as a sequence of spatio-temporal points, capturing the movement of a mobile object (e.g., a person or vehicle) and enabling various applications (Jeong et al., 2014; LOU et al., 2021), such as similarity search (Li et al., 2018), and transportation mode classification (Hu et al., 2024). Traditional approaches often rely on manually extracted features for these analyses (Wang et al., 2020), overlooking hidden correlations between trajectories and thereby limit performance. Recently, ***trajectory representation learning*** has emerged (Chen et al., 2021; Fu & Lee, 2020; Jiang et al., 2023a), aiming to transform high-dimensional trajectories into low-dimensional vectors (i.e., trajectory embeddings) that retain essential information from the original data and capture hidden features. These vectors are then fed into a range of trajectory analysis tasks.

Existing trajectory representation learning studies can be divided into two categories: (i) **free-space settings** and (ii) **road-network settings**. In free-space settings, early studies (Fang et al., 2021; Li et al., 2018) treat trajectories as pure point sequences, disregarding road network constraints. Thus, they typically apply sequential models like LSTMs and RNNs to capture the spatio-temporal dependencies in trajectory data. Since moving objects, such as people and vehicles, are constrained by road networks, road-network-based trajectory representation learning methods have been developed. These approaches (Fang et al., 2022; Fu & Lee, 2020; Han et al., 2021; Yao et al., 2022) typically begin by learning embeddings for road segments using graph neural networks (GNNs) with road network graphs as input. Subsequently, hidden spatio-temporal relations can be captured by feeding the road segment embeddings into sequential models, which are trained on task-specific objectives. Recently, state-of-the-art methods, including START (Jiang et al., 2023a) and JGRM (Ma et al., 2024), have adopted self-supervised learning paradigms to improve the generalization of trajectory representation learning across various tasks. More detailed works can be found in Appendix A. However, there are still some unsolved challenges in devel-

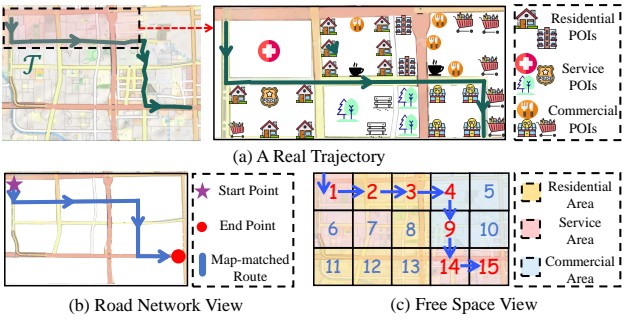

(a) A Real Trajectory

(b) Road Network View

(c) Free Space View

*Figure 1.* Free Space View vs. Road Network View

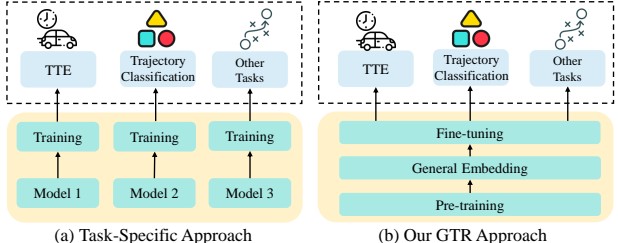

(a) Task-Specific Approach     (b) Our GTR Approach

*Figure 2.* Task-Specific Approach vs. Our General Approach

oping an effective trajectory representation learning model.

***C1: Limitation of single-view representation.*** As mentioned, existing representation learning methods generally model trajectory data from a single perspective—either a free space view or a road network view. However, trajectory data encompasses complex spatio-temporal semantic information, while many essential spatio-temporal features and semantic nuances required for downstream tasks cannot be fully captured from a single view. Fig. 1(a) depicts a real trajectory $\mathcal{T}$ on the road network, along with the complex surrounding environment. Fig. 1(b) presents a representation of $\mathcal{T}$ from the road-network view, which captures only its topological structure while neglecting the regional semantics of the traversed areas. As a result, the trajectory embeddings learned in this manner may fail to encode the latent semantic meanings of the passed road segments or regions, potentially undermining downstream tasks such as semantic-aware trajectory similarity searches. In contrast, Fig. 1(c) illustrates the trajectory's representation in free-space view using a grid partitioning method (Li et al., 2018). This example indicates that relying on a single view may fail to capture the multi-faceted features of trajectories. Although recent studies (Lin et al., 2023; Ma et al., 2024; Yi et al., 2024) have incorporated multi-dimensional information for trajectory learning, they differ from our setting, as they encode the road segments with GPS points. In contrast, we encode the road segments and grids with the latent semantic features, enabling more fine-grained spatio-temporal learning, as proved by experiments. Overall, how to collaboratively integrate multiple views for trajectory representation learning is a challenge.

***C2: Limitation of multitasking.*** As aforementioned, quite a few approaches (Fang et al., 2023; Jiang et al., 2023b; Si et al., 2023) are designed for specific trajectory analysis tasks, which limits their generalizability across different applications. As shown in Fig 2(a), the task-specific approaches require training separate models for each task, resulting in substantial development costs. In contrast, we aim to propose a general approach, illustrated in Fig. 2(b), to support a wide range of downstream tasks. While the self-supervised learning paradigm of pre-training and fine-

tuning has achieved great success in the representation learning field (Fu & Lee, 2020), its direct utilization to trajectory representation learning (Jiang et al., 2023a; Ma et al., 2024) may hinder the model's robustness and generalization. According to Table 9, even the state-of-the-art methods (Chen et al., 2021; Fu & Lee, 2020; Jiang et al., 2023a; Ma et al., 2024; Yang et al., 2021b) only address a limited subset of trajectory analysis tasks. This limitation arises due to conflicting correlations between different tasks. Therefore, how to balance the learning of spatial and temporal features to automatically adapt to various trajectory analysis tasks remains an unaddressed challenge.

***C3: Lack of support for model update.*** Trajectory data exhibit strong dynamic characteristics, particularly in urban areas where large volumes of trajectories are continuously generated (Chen et al., 2024). This constant influx of new data creates an evolving context, as trajectory movement patterns continuously adapt to changing traffic conditions. Consequently, it is crucial to continuously learn the latest spatio-temporal features from new trajectories to maintain an accurate and up-to-date representation learning model. However, as shown in Table 9, none of the state-of-the-art methods currently support model updating, limiting their effectiveness in dynamic environments where data patterns are constantly shifting. Achieving online model updates presents a significant challenge, as streaming trajectories exhibit complex spatio-temporal correlations that are computationally intensive to extract in real time.

**Contributions.** To address the challenges above, we propose GTR, a General, multi-view, and dynamic Trajectory Representation learning framework, designed to generate robust embeddings that support various downstream trajectory analysis tasks. To track challenge ***C1***, we introduce an effective Multi-View Encoder (MVE), which encodes the original trajectories from both free-space and road-network perspectives, integrating semantic regional and road topology information to capture sufficient spatio-temporal features. To overcome challenge ***C2***, we propose Spatio-Temporal fusion Pre-training (STP) based on Transformer, where we devise a Spatio-Temporal Mixture of Expert (ST-MoE) module to learn and adapt distinct spatio-temporal features required by various tasks in a data-driven manner, providing a dynamic approach to integrate spatio-temporal features. In the fine-tuning stage, we provide a suite of tuning methods

Table 1. Notations and Descriptions

| Notation | Description |
|---|---|
| $\mathcal{T}$ | GPS trajectory. |
| $\mathcal{T}^g$ | Grid-constrained trajectory. |
| $\mathcal{T}^r$ | Road-network-constrained trajectory. |
| $\mathcal{G}$ | Grid cells. |
| $G$ | Road network. |
| $D^{\mathcal{T}}$ | Road trajectory dataset. |
| $Z_R$ | Road representation. |
| $Z_G$ | Grid representation. |
| $Z_P$ | Position representation. |
| $Z_T$ | Temporal representation. |
| $Z_S$ | Spatial representation. |
| $h$ | Trajectory generalized representation. |

for diverse downstream tasks. To contend with challenge **C3**, we propose an efficient Online Frozen-hot Updating (OFU) strategy by selectively freezing the parameters of the Transformer encoder during model updating. Moreover, to enhance the interpretability of the trajectory representative model, we calculate the attention values to identify crucial information that influences the training process. Based on that, visualization processes illustrate the learning procedure and provide optimization guidelines. Finally, we conduct extensive experiments on two real-world datasets to demonstrate that GTR outperforms 15 state-of-the-art baselines across various trajectory analyses.

## 2. Preliminaries

**Notations Table.** We present the frequently used notations and descriptions in this paper, as listed in Table 1.

**Definition 2.1** (**GPS Trajectory**). A GPS trajectory $\mathcal{T}$ is denoted as a sequence of GPS spatio-temporal points, i.e., $\mathcal{T} = \langle p_i | (1 \leq i \leq L) \rangle$, where each point $p_i = (lon_i, lat_i, t_i)$ contains longitude, latitude, and observed timestamp, $p_i$ is the $i$-th point of $\mathcal{T}$, $L$ denotes the length of $\mathcal{T}$.

**Definition 2.2** (**Road Network**). A road network is denoted as a directed graph $G = (V, E, A)$. $V$ is the set of graph vertices, where each $v_i \in V$ denotes a road segment. $E \subseteq V \times V$ is a set of graph edges, where each $e_{ij} = (v_i, v_j) \in E$ denotes an intersection between $v_i$ and $v_j$. $A \in \mathbb{R}^{|V| \times |V|}$ denotes the binary adjacency matrix of graph $G$.

**Definition 2.3** (**Road-network Constrained Trajectory**). A road-network constrained trajectory $\mathcal{T}^r$ is a time-ordered sequence of adjacency road segments, i.e., $\mathcal{T}^r = \langle (v_i, t_i^r) | (1 \leq i \leq L_r, v_i \in V) \rangle$, where $t_i^r$ is the visit timestamp for $v_i$, and $L_r$ is the length of $\mathcal{T}^r$.

Following previous free space trajectory representation learning (Fang et al., 2021; Li et al., 2018), we partition the free space into $w_1 \times w_2$ grid cells and assign each GPS point to the grid cell that contains it. All of these grids make up a set $\mathcal{G} = \langle g_i | (1 \leq i \leq w_1 \times w_2) \rangle$. Each GPS trajectory

is then mapped to a grid-constrained trajectory, offering auxiliary spatial information to enhance feature extraction within the road network context.

**Definition 2.4** (**Grid Constrained Trajectory**). A grid constrained trajectory $\mathcal{T}^g$ is a time-ordered sequence of grid cells, i.e., $\mathcal{T}^g = \langle (g_i, t_i^g) | (1 \leq i \leq L_g, g_i \in \mathcal{G}) \rangle$, where $\mathcal{G}$ is a set of grid cells, $t_i^g$ is the visit timestamp for $g_i$, and $L_g$ is the length of $\mathcal{T}^g$.

**Problem Statement.** For a road network $G$ and a trajectory dataset $D^{\mathcal{T}} = \langle \mathcal{T}_i | (1 \leq i \leq |D^{\mathcal{T}}|) \rangle$, our goal is to learn a generalized representation $h_i$ for each trajectory $\mathcal{T}_i$ ($\mathcal{T}_i \in D^{\mathcal{T}}$). This representation should capture essential spatiotemporal and semantic features to effectively support multiple downstream tasks, such as similarity search, imputation, generation, classification, simplification, and travel time estimation.

## 3. Methodology

**Framework Overview.** The GTR framework is illustrated in Fig. 3, comprising three key components: the **Multi-View Encoder** (MVE), the **Spatio-Temporal fusion Pre-training** (STP), and the **Online Frozen-hot Updating** (OFU). Together, these components are designed to generate generalized representations for road network-constrained trajectories. In the sequel, we detail each component in order.

### 3.1. Multi-View Encoder (MVE)

**Design Motivation.** Previous works (Fu & Lee, 2020; Jiang et al., 2023a) mainly focus on road networks with static semantics, ignoring the spatial features of free-space view, such as area function characteristics (cf. Fig. 1). Moreover, the free-space view provides a coarse-grained view of the trajectory, aiding the model in learning the trajectory's overall trend. Therefore, we design an MVE module to encode GPS trajectories, extracting multi-view representations and spatio-temporal features by four embedding procedures.

**Preparation.** Given a road network $G$ and a raw trajectory $\mathcal{T}$, we first encode $\mathcal{T}$ into road network-constrained and grid-constrained trajectories. Specifically, any map-matching algorithms (Newson & Krumm, 2009; Ruan et al., 2018; Yang & Gidofalvi, 2018) can be employed to establish correspondences between the GPS points of $\mathcal{T}$ and the road segments in $G$, ensuring road network connectivity constraints. This process generates the road network-constrained trajectory $\mathcal{T}^r$ (cf. Definition 2.3), denoted as $\mathcal{T} \rightarrow \mathcal{T}^r$. Simultaneously, the space is partitioned into grid cells, with each grid classified based on its contained POIs from OpenStreetMap[1]. As a result, the input $\mathcal{T}$ is also represented as a grid-constrained trajectory $\mathcal{T}^g$ (cf. Definition 2.4), denoted as $\mathcal{T} \rightarrow \mathcal{T}^g$. Using $\mathcal{T}^r$ and $\mathcal{T}^g$, we extract

---

[1]https://www.openstreetmap.org/

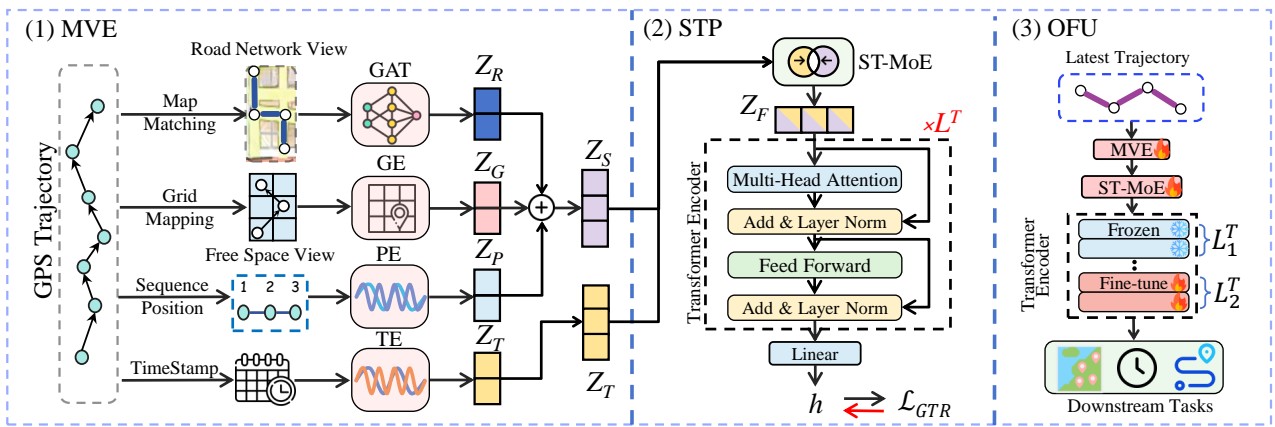

*Figure 3.* The Overall Framework of GTR

semantic and spatio-temporal features via four embedding methods as follows:

i) **Road embedding.** We begin by collecting the features and semantic information of road segments in $G$, such as speed limits, road types, and road lengths, collected from OpenStreetMap, denoted as $F_v = [f_1, f_2, \cdots, f_{|V|}]$. Next, Graph Attention Network (GAT) layers are applied to encode $F_v$ into the road embedding $Z_R$. This process computes an attention coefficient matrix to capture the influence among features. The operation at the $l$-th GAT layer is mathematically defined as follows:

$$AE_{ij} = \mathbf{a}^\top (\mathbf{W}_{v_0} f_i \| \mathbf{W}_{v_1} f_j) \qquad (1)$$

Here, $f_i$ represents the extracted features or semantic information of $v_i$ ($v_i \in V$). $AE_{ij}$ denotes the attention coefficient between $f_i$ and $f_j$, which depicts the influence among features. $\mathbf{W}_{v_0}$, $\mathbf{W}_{v_1}$ and $\mathbf{a}^\top$ denote the learnable parameters. Next, the attention value is calculated and normalized to aggregate the vertex information of $G$:

$$\alpha_{ij} = \frac{\exp(\text{LeakyReLU}(AE_{ij}))}{\sum_{k \in \mathcal{N}_i} \exp(\text{LeakyReLU}(AE_{ik}))}, \qquad (2)$$

where LeakyReLU denotes the activation function. $\alpha_{ij}$ denotes the attention value between $v_i$ and $v_j$, and $\mathcal{N}_i$ represents the set of all neighbors of $v_i$. Finally, the features $f_i$ ($f_i \in F_v$) can be represented as:

$$f_i^{l+1} = \|_{k=1}^{a'} \text{ELU} \left( \sum_{j \in \mathcal{N}_i} \alpha_{ij}^{(k)} \mathbf{W}_{v_3}^{(k)} f_j^{(l)} \right) \qquad (3)$$

Here, ELU is the Exponential Linear Unit activation function (Veličković et al., 2017), $\|$ represents the "CONCAT" operation. $\alpha_{ij}^{(k)}$ denotes the attention value computed by the $k$-th attention head, and $a'$ is the number of the attention heads. $\mathbf{W}_{v_3}^{(k)}$ is the weight matrix of the corresponding linear transformation in layer $l$. The final road embedding can be represented as $Z_R = [f_{v_1}, f_{v_2}, \cdots, f_{v|Lr|}]$.

ii) **Grid embedding with POIs.** We aim to transfer the each grid constrained trajectory of $\mathcal{T}^g$ into a grid embedding $h_g$.

Specifically, we use an embedding vector to represent each grid cell, then map the trajectories to the corresponding embedding vectors. The process is as below:

$$h_{gi} = \text{GE}(\mathcal{T}_i^g) + \text{POIE}(c_i^{poi}) \; (1 \le i \le L_g) \qquad (4)$$

Here, GE, POIE construct the embedding vectors for grid-constrained trajectories $\mathcal{T}^g$ and grid's latent type $c^{poi}$. Our POIs extraction method can be found in Appendix B.6. It is worth mentioning that combining grid embedding with POI information provides auxiliary insights, capturing features overlooked by road embeddings and enhancing the accuracy of trajectory representations. The grid embedding can be represented as $Z_G = [h_{g1}, h_{g2}, \cdots, h_{gL_g}]$.

iii) **Position Embedding.** To model the sequential dependencies within a trajectory, we employ position embeddings (Devlin et al., 2019) to encode the order of the input trajectory $\mathcal{T}$. These embeddings are generated using sine and cosine functions (Devlin et al., 2019) as follows:

$$\text{PE}_{(pos,2i)} = \sin\left(\frac{pos}{10000^{2i/d}}\right), \; \text{PE}_{(pos,2i+1)} = \cos\left(\frac{pos}{10000^{2i/d}}\right), \qquad (5)$$

where PE denotes the position embedding, and $pos$ denotes the sequence position in the trajectory $\mathcal{T}$. $d$ denotes the embedding size of GTR. With the position embedding, we can get the position embedding $h_{pi} = [\text{PE}_{(1,0)}, ..., \text{PE}_{(1,d-1)}, ..., \text{PE}_{(|\mathcal{T}_i|,0)}, ..., \text{PE}_{(|\mathcal{T}_i|,d-1)}]$, and the position embedding can be represented as $Z_P = [h_{p1}, h_{p2}, ..., h_{pL}]$.

iv) **Time Embedding.** We aim to capture multi-granularity temporal features for effective trajectory representation, denoted as $h_t$. Specifically, we provide three temporal features for each timestamp of $\mathcal{T}$: minutes, weeks, and years. By using three embedding vectors to extract the temporal patterns of minutes, weeks, and years, we obtain the corresponding embedding vectors for each timestamp, as defined below:

$$h_{ti} = \text{TE}_m(f_t(t_i)) + \text{TE}_w(f_t(t_i)) + \text{TE}_y(f_t(t_i)) \qquad (6)$$

Then, we can obtain the temporal feature representation $Z_T = [h_{t1}, h_{t2}, ..., h_{tL}]$. $\text{TE}_m$, $\text{TE}_w$, and $\text{TE}_y$ construct the

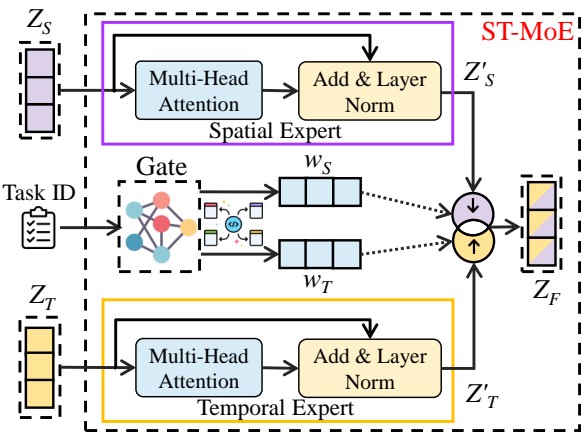

*Figure 4.* The ST-MoE Module

embedding vectors for minute, week, and year representations, respectively. We use the function $f_t$ to transform timestamps into index ranges: [0, 1440] for minutes, [0, 7] for weeks, and [0, 366] for years.

Finally, we combine the grid embedding $Z_G$, road embedding $Z_R$, and position embedding $Z_P$ to derive the spatial feature representation, computed as follows:

$$Z_S = Z_R + Z_G + Z_P \qquad (7)$$

### 3.2. Spatio-Temporal Fusion Pre-training (STP)

**Design Motivation.** Recent works (Jiang et al., 2023a; Lin et al., 2023) mainly focus on the subset of the trajectory downstream tasks, such as TTE, trajectory classification, and most similar trajectory search. However, the pre-training task is not usually suit for all downstream tasks, due to the limitation of the invariant feature combination. Therefore, we design an STP module to solve this gap, aiming to develop a dynamic trajectory representation learning model, enabling the generation of general trajectory representations that support multiple downstream tasks. The process comprises three stages: (i) spatio-temporal feature fusion, (ii) pre-training, and (iii) fine-tuning.

#### 3.2.1. SPATIO-TEMPORAL FUSION.

Different downstream tasks require varying proportions of spatio-temporal features, necessitating dynamic adjustments. Inspired by the effectiveness of Mixture of Experts (MoE) methods (Cai et al., 2024) in feature fusion across diverse domains, we propose the ST-MoE module, specifically designed to dynamically integrate spatio-temporal features. An overview of the ST-MoE module is shown in Fig. 4. Specifically, this module includes two experts: a spatial expert and a temporal expert. First, the spatial feature representation $Z_S$ and temporal feature representation $Z_T$ are fed into their respective experts. The spatial expert processes $Z_S$ using a multi-head attention layer (Vaswani et al., 2017) followed by a residual connection layer. The temporal expert follows the same process. The detailed procedure is as

follows:

$$Z'_S = LayerNorm(Z_S + Dropout(\mathbf{MultiHead}(Z_S))), \quad (8)$$

$$\mathbf{MultiHead}(Z_S) = \{head_1, \cdots, head_{L_{head}}\} \cdot \mathbf{W}^O_{Z_S}, \quad (9)$$

$$head_i = \mathbf{Attention}(Z_S \mathbf{W}^Q_{Z_{Si}}, Z_S \mathbf{W}^K_{h_{Si}}, Z_S \mathbf{W}^V_{Z_{Si}}), \quad (10)$$

where $L_{head}$ donates the number of the attention head, $\mathbf{W}^Q_{Z_{Si}}, \mathbf{W}^K_{Z_{Si}}, \mathbf{W}^V_{Z_{Si}}, \mathbf{W}^O_{Z_S}$ are learnable parameters, *LayerNorm* donates the Layer Normalization (Lei Ba et al., 2016). We obtain $Z'_T$ in the same way, which process is omitted due to space limitations.

Next, we design a gating network to generate spatio-temporal weights for the experts. For different tasks, the task ID $x_{task}$ is used to derive the task representation through a fully connected layer. The weights are then computed using a *Softmax* function. The process is defined as follows:

$$w_S, w_T = Softmax(FC(x_{task})), \qquad (11)$$

where $w_S$ is the spatial weight, $w_T$ is the temporal weight, *Softmax* is the *Softmax* activation function, and *FC* is the fully connected layer.

Finally, we obtain the fusion road representation through a residual connection and normalization layer as follows:

$$Z'_F = Z'_S \cdot w_S + Z'_T \cdot w_T, \qquad (12)$$

$$Z_F = LayerNorm(Z'_F + Dropout(Z'_F)), \qquad (13)$$

where $Z_F$ is the fusion representation, which is the input of the next pre-training stage.

#### 3.2.2. PRE-TRAINING.

Leveraging the Transformer's powerful ability to capture long-distance dependencies within sequences and its efficiency in parallel computing (Yang et al., 2023), we adopt a Transformer encoder for pre-training to learn general trajectory representations through carefully designed pre-training tasks. Specifically, we utilize the Masked Language Model (MLM) as the pre-training task for trajectory representation due to its proven effectiveness in handling sequential data (Devlin et al., 2019). In this approach, MLM randomly selects tokens from the sequence, masks them, and trains the Transformer encoder to predict the masked tokens, thereby capturing rich contextual information.

However, directly applying this technique to road network-constrained trajectory representation introduces limitations. This is, the adjacency of road segments in trajectories makes it relatively simple for the Transformer encoder to predict masked tokens. Consequently, treating individual road segments as sequence tokens results in a pre-trained model that lacks the complexity needed to support advanced downstream tasks. To overcome this limitation, we propose a new span masking method, which masks consecutive road segments (i.e., sub-trajectories) instead of individual segments. This method consists of three main steps.

**Step 1:** For the road network-constrained trajectory dataset $D^{\mathcal{T}^r}$, we apply data augmentation techniques to enrich the training data and provide additional information for model optimization. These techniques include sub-trajectory selecting and road drifts, which can be found in Appendix B.5.

**Step 2:** Given a road network constrained trajectory $\mathcal{T}^r$ to be masked, we randomly select 30% of road segments from $\mathcal{T}^r$ and mask them by replacing them. Next, we obtain the feature representation $Z_F$ of the masked trajectory (cf. Eq. 12) and compute the weighted representations $h$ based on $Z_F$ using a multi-head attention layer (Vaswani et al., 2017). The multi-head attention layer captures information from multiple views (e.g., grid view and road network view), thereby improving the accuracy of trajectory representation learning. Firstly, We obtain the query, key and value vectors $\mathbf{Q}_{Z_F} = Z_F \mathbf{W}_{Z_F}^Q$, $\mathbf{K}_{Z_F} = Z_F \mathbf{W}_{Z_F}^K$, $\mathbf{V}_{Z_F} = Z_F \mathbf{W}_{Z_F}^V$, where $\mathbf{W}_{Z_F}^Q$, $\mathbf{W}_{Z_F}^K$, $\mathbf{W}_{Z_F}^V$ are learnable parameters. This step is defined as follows:

$$h' = \mathbf{MultiHead}(\mathbf{Q}_{Z_F}, \mathbf{K}_{Z_F}, \mathbf{V}_{Z_F}), \tag{14}$$

$$h = (\mathbf{ReLU}(h'\mathbf{W}_0 + \mathbf{b}_0))\mathbf{W}_1 + \mathbf{b}_1, \tag{15}$$

where $\mathbf{Q}_{Z_F}$, $\mathbf{K}_{Z_F}$, and $\mathbf{V}_{Z_F}$ represent the query, key and value vectors obtained by linear transformation for $h$, respectively. $\mathbf{ReLU}$ is the activation function. $\mathbf{W}_0, \mathbf{W}_1, \mathbf{b}_0$, and $\mathbf{b}_1$ denote the learnable parameters.

**Step 3:** We train the model to predict the masked road segments, using the cross-entropy loss (Mao et al., 2023):

$$\hat{y} = Softmax(\mathbf{W}_2 h + \mathbf{b}_2), \ \ \mathcal{L}_c = -\sum_{i=1}^{N_{mask}} \sum_{c=1}^{C} y_{i,c} \log(\hat{y}_{i,c}), \tag{16}$$

where *Softmax* is the activation function. $\mathbf{W}_2$ and $\mathbf{b}_2$ denote the learnable parameters. $N_{mask}$ denotes the number of masked tokens, and $C$ represents the size of the sub-trajectories that are masked, $\hat{y}_{i,c}$ is the model prediction.

**Training Optimization.** To further improve the accuracy of the pre-trained model, we introduce an additional pre-training task that constructs trajectory triplets to capture the relationships among them. Specifically, for each $\mathcal{T}^r$, we designate it as an anchor trajectory $\mathcal{T}_a^r$, and extract its sub-trajectory as the positive sample $\mathcal{T}_p^r$. Unlike previous methods (Jiang et al., 2023a; Ma et al., 2024), which construct negative samples directly from the original trajectory data and struggle to distinguish dissimilar trajectories, we provide negative samples of varying difficulty levels to learn more nuanced differences between trajectories. First, we randomly select several trajectories $\mathcal{T}^r$ from the trajectory Dataset $D^r$ as simple samples. Second, we employ a Variational Autoencoder (VAE) (Doersch, 2016) to generate more challenging negative samples. The reconstruction task and Kullback-Leibler (KL) divergence are employed during training to generate negative samples for the anchor. This

results in the trajectory triplet ($\mathcal{T}_a^r$, $\mathcal{T}_p^r$, $\mathcal{T}_n^r$) and its corresponding representation triplets ($h^a$, $h^p$, $h^n$) (cf. Eq. 15). Using these triplets as training samples, we train the Transformer encoder. During training, the model keeps the anchor trajectory closer to the positive sample and farther from the negative sample. Note that, the training procedure is self-supervised, as the triplet samples are generated from the trajectories themselves. The loss function is defined below:

$$\mathcal{L}_t = \sum_{i}^{N} \left[ \|h_i^a - h_i^p\|_2^2 - \|h_i^a - h_i^n\|_2^2 + \tau \right]_+, \tag{17}$$

where $N$ is the number of training samples (i.e., triplets). $\tau$ represents the parameter that defines the minimum margin required between positive and negative samples.

Overall, we aim to obtain a pre-trained model by training on the two tasks described above, which is handled by the loss function defined as follows:

$$\mathcal{L}_{GTR} = \beta * \mathcal{L}_c + (1 - \beta) * \mathcal{L}_t, \tag{18}$$

where $\beta$ is an adjusting parameter to balance the influence of two pre-train tasks (i.e., MLM and Triplet Training).

### 3.2.3. FINE-TUNING.

With the pre-trained model described above, we perform fine-tuning for each downstream task to achieve superior performance. Unlike other methods (Fu & Lee, 2020; Jiang et al., 2023a; Ma et al., 2024), which apply the same fine-tuning approach across all tasks, we propose tailored fine-tuning strategies for each downstream task. Due to the limited space, we only introduce the Travel Time Estimation task here, more fine-tuning methods refer to Appendix B.1.

**Travel Time Estimation.** The goal of TTE is to estimate the travel time for a moving object from a start point to a destination. To achieve this, we construct a regression model to estimate the travel time based on a fully connected layer of neural networks. Then, we use the Huber loss (Shi et al., 2023) for fine-tuning, shown as below:

$$\mathcal{L}_{regression} = \begin{cases} \frac{1}{2}(y - \hat{y})^2 & \text{for} |y - \hat{y}| \leq \delta \\ \delta \left( |y - \hat{y}| - \frac{1}{2}\delta \right) & \text{otherwise} \end{cases}, \tag{19}$$

where $\delta$ is a preset threshold, we set $\delta = 1$ in our work. $\hat{y}$ is the predicted value by the model, and $y$ is the true label.

### 3.3. Online Frozen-Hot Updating (OFU)

Previous approaches (Chen et al., 2024; Ma et al., 2024) often fail to leverage the real-time capabilities of trajectory data, reducing their effectiveness in dynamic environments where traffic conditions are constantly evolving. To address this limitation, we propose an online frozen-hot updating strategy. Additionally, we introduce model interpretation methods to explain the rationale behind the model's outputs, facilitating optimization through well-founded approaches, which can be found in Appendix B.3 due to the limited space.

**Online Updating.** We consider the latest trajectory data for model updating, which is continuously collected. However, retraining the model using the latest trajectory data as training samples would delay downstream tasks, resulting in low efficiency. Therefore, it is essential to update the model while ensuring that downstream tasks continue to perform effectively. Additionally, historical trajectories must be preserved during model updates, as they contain vital information for trajectory analysis.

With this in mind, we propose an incremental online updating strategy. Specifically, first, for the Transformer encoder with $L^T$ layers, we divide the layers into two parts: $L_1^T$ and $L_2^T$. Second, we freeze $L_1^T$ to preserve the information of historical trajectories for downstream tasks, while continuing to pre-train $L_2^T$ using the latest trajectories as training samples. In this way, the model can perform trajectory analysis tasks and update simultaneously, achieving online optimization. Third, we update the model as new trajectories flow in, iteratively executing the second step. It is worth mentioning that the impact of historical and recent trajectories can be managed by adjusting the number of layers in $L_1^T$ (denoted as $\epsilon$) or $L_2^T$ (denoted as $L^T - \epsilon$). In this paper, we set $\epsilon$ to be half of $L^T$ to balance the influence of historical and recent trajectory information.

Note that, the OFU strategy is theoretically grounded in incremental learning (Wang et al., 2024). We combine layer-wise parameter freezing with Lyapunov stability analysis to ensure robust adaptation while preventing catastrophic forgetting. Specifically, we freeze the first $L$ layers and update the last $N - L$ layers. The objective optimization function is defined below:

$$\min_{\theta^{L+1:N}} \mathbb{E}_{(x,y)} \sim \mathcal{D}_{\text{new}}[\ell(f_{\theta_{\text{pre}}^{1:L}}(x), y; \theta^{L+1:N})] \quad (20)$$

With frozen lower-layer parameters ($\nabla_{\theta^{1:L}} \ell = 0$), old features remain unchanged, which guarantees that the old features are not forgotten. The updating process can be modeled as a dynamic system: $\theta_{t+1}^{L+1:N} = \theta_t^{L+1:N} - \eta_t g_t$, with a Lyapunov function: $V(\theta) = \mathcal{L}_{\text{new}}(\theta) + \gamma \|\theta^{1:L} - \theta_{\text{pre}}^{1:L}\|^2$. As $\gamma \to \infty$, the system satifies: $\mathbb{E}[V(\theta_{t+1})] \leq \mathbb{E}[V(\theta_t)] - \eta_t \|\nabla \mathcal{L}_{\text{new}}(\theta_t)\|^2$. The monotonic decrease of $V(\theta)$ ensures stable updates. Freezing lower layers prevents cascading perturbations, balancing new feature learning with old feature retention.

### 3.4. The Training of GTR

The training process of GTR is provided in Appendix B.2 with the complexity analysis in Appendix B.4.

## 4. Experiments

In this section, we conduct a series of experiments on two real-world datasets to evaluate the performance of GTR, which are summarized to answer the following questions.

- **RQ1:** *How does GTR perform compared to the state-of-the-art models on supporting multiple tasks?*
- **RQ2:** *How effective is our online updating strategy for model training?*
- **RQ3:** *How do the individual modules in GTR contribute to the model performance?*
- **RQ4:** *How does GTR interpret model training?*
- **RQ5:** *What is the scalability of our GTR?*

### 4.1. Experimental Settings

*Table 2.* Dataset Statistics

| Datasets | # segments | # trajectories | Avg. Length | Avg. Time |
|---|---|---|---|---|
| Porto | 44,641 | 774,262 | 41 | 9.901 |
| Beijing | 40,305 | 889,306 | 27 | 12.833 |

**Datasets.** We evaluate GTR on two real-world trajectory datasets: (i) Porto[2] contains 774,262 GPS trajectories collected in Porto, Portugal, from 2013/07/01 to 2014/07/01. (ii) Beijing [3] contains 889,306 GPS trajectories collected in Beijing, China, from 2015/11/01 to 2015/11/30. Detailed information is summarized in Table 2. The Avg.Length refers to the average number of trajectory points.

**Baselines Description.** We compare GTR with 15 most common state-of-the-art methods among six tasks.

- **TremBR (Fu & Lee, 2020)** constructs a RNN-based seq2seq model for representation leanring.
- **PIM (Yang et al., 2021b)** combines node2vec and LSTM encoder to generate trajectory representations.
- **Toast (Chen et al., 2021)** utilizes node2vec model and trains a Transformer encoder to represent trajectories.
- **START (Jiang et al., 2023a)** trains a time-aware encoder with a GAT that considers transitions in the road network.
- **LightPath (Yang et al., 2023)** trains a sparse path encoder for path reconstruction and cross-view network contrast.
- **TS-TrajGen (Jiang et al., 2023b)** builds an A* algorithm based generator within a generative adversarial network.
- **SeqGAN (Yu et al., 2017)** utilizes GANs combined with Seq2Seq models for trajectory representation.
- **Trajbert (Si et al., 2023)** trains a Transformer encoder by using a spatial-temporal loss for trajectory recovery.
- **Bi-STDDP (Xi et al., 2019)** is designed to integrate bidirectional spatio-temporal dependencies and users' dynamic preferences, identifying missing POIs.
- **EB-OTS (Wang et al., 2021)** proposes a multi-agent for online trajectory simplification.
- **S3 (Fang et al., 2023)** constructs a lightweight framework using two seq2seq models to simplify trajectories.
- **JGRM (Ma et al., 2024)** designs a novel representation model that jointly encodes GPS and routes through a Transformer.
- **AttnMove (Xia et al., 2021)** recovers dense trajectories

[2]https://www.kaggle.com/c/pkdd-15-predict-taxi-service-trajectory-i
[3]https://github.com/aptx1231/START

by inferring unobserved locations through a multi-layer attention-based neural network.

- **Mtrajrec (Ren et al., 2021)** integrates a GRU model with an attention mechanism to enhance trajectory recovery.
- **ControlTraj (Zhu et al., 2024)** leverages a diffusion model to efficiently generate trajectories.

**Implements.** All experiments are conducted on CentOS 7 with an NVIDIA A40 GPU. We run GTR on PyTorch 1.13.1. For GTR, we set the embedding size as 768, the mask ratio for pre-training tasks as 30 %, and the dropout value as 0.2. Moreover, we perform pre-training and fine-tuning of GTR using the AdamW (Loshchilov & Hutter, 2019) optimizer, and the value of the balancing parameter $\beta$ in the overall loss function is set to 0.7. Our dataset is split into training, validation, and test sets with a ratio of 0.8, 0.1, and 0.1. More detailed settings refer to Appendix C.1.

### 4.2. Performance Evaluation (*RQ1 & RQ2*)

In this section, we first evaluate the accuracy of all methods on six usual downstream tasks, including trajectory similarity computation, trajectory simplification, trajectory imputation, travel time estimation, trajectory classification, and trajectory generation. Note that, we only evaluate the baselines' performance on their specific tasks. Next, we incrementally update the training model online to obtain the updated model GTR*, and compute the accuracy to verify the online updating strategy. To process newly arrived trajectory data, we employ the validation set to simulate real-world online/streaming scenarios. The model undergoes single-epoch incremental model updates.

**Trajectory Similarity Computation:** we perform Top-$k$ similarity search by computing Mean Rank (MR), HR@1, HR@5 (Hu et al., 2023) for Trembr, PIM, Toast, START, JGRM, LightPath, and GTR (lower MR and higher HR@1 and HR@5 indicate the higher performance). Note that, we apply the detour method in JCLRNT (Mao et al., 2022) to generate the ground truth. The results are reported in Table 3. Further, we present a case study, shown in Appendix C.7.

*Table 3.* Evaluation on Top-$k$ Similarity Computation Task

| | Beijing | | | Porto | | |
|---|---|---|---|---|---|---|
| Methods | MR | HR@1 | HR@5 | MR | HR@1 | HR@5 |
| Trembr | 2.4894 | 0.6070 | 0.9084 | 8.6536 | 0.6226 | 0.8424 |
| PIM | 4.8008 | 0.9462 | 0.9816 | 27.0137 | 0.6064 | 0.7928 |
| Toast | 1.6656 | 0.9036 | 0.9850 | 14.6700 | 0.7554 | 0.8716 |
| START | 1.3870 | 0.8968 | 0.9888 | 1.0476 | 0.9722 | 0.9984 |
| LightPath | 1.1116 | 0.9312 | 0.9982 | 4.1272 | 0.7308 | 0.8808 |
| JGRM | 1.5946 | 0.9720 | 0.9938 | 5.9172 | 0.2326 | 0.6634 |
| GTR | **1.0130** | **0.9906** | **0.9996** | **1.0028** | **0.9974** | **0.9999** |

**Trajectory Simplification:** we compute Perpendicular Distance Error (PED) (Fang et al., 2023) of S3, EB-OTS, GTR, and GTR* for performance evaluation due to the limited space. Note that, PED computes the shortest perpendicular distance between a deleted point $p_i = (x_i, y_i)$ and the line segment connecting its neighboring points $p_s = (x_s, y_s)$

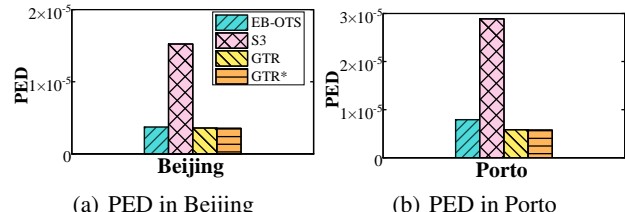

| (a) PED in Beijing | (b) PED in Porto |
|---|---|

*Figure 5.* Evaluation on Trajectory Simplification Task

and $p_t = (x_t, y_t)$. Lower PED indicates lower compression errors. The results are shown in Fig. 5.

**Trajectory Imputation:** we assess the accuracy performance using Recall@3, Recall@5, and Mean Accuracy Percent (MAP) for Bi-STDDP, AttnMove, Trajbert, MtrajRec, GTR, and GTR*. Recall@x evaluates the model's ability to recover masked tokens by checking whether the ground truth appears in the top-x predicted candidates. Specifically, if the true value is contained within the top-x ranked predictions, Recall@x is assigned 1 for that token; otherwise, it is assigned 0. The final metric is computed by averaging these binary outcomes across all masked tokens in the evaluation set. The MAP represents the probability of the precision, and higher Recall@x and MAP indicate higher performance. The results are shown in Table 4.

*Table 4.* Evaluation on Trajectory Imputation Task

| | Beijing | | | Porto | | |
|---|---|---|---|---|---|---|
| Methods | Recall@3 | Recall@5 | MAP | Recall@3 | Recall@5 | MAP |
| Bi-STDDP | 0.97874 | 0.98338 | 0.82692 | 0.61926 | 0.74273 | 0.33584 |
| AttnMove | 0.94556 | 0.96151 | 0.78721 | 0.57879 | 0.69355 | 0.31801 |
| Trajbert | 0.93476 | 0.95059 | 0.85206 | 0.65174 | 0.76151 | 0.37843 |
| MtrajRec | 0.94840 | 0.97640 | 0.87650 | 0.68170 | 0.73090 | 0.40920 |
| GTR | 0.99406 | 0.99521 | 0.98652 | 0.86465 | 0.93331 | 0.60344 |
| GTR* | **0.99418** | **0.99531** | **0.98657** | **0.86525** | **0.93354** | **0.60538** |

**Travel Time Estimation:** we compare Trembr, Toast, START, PIM, LightPath, and JGRM with GTR and GTR*, by utilizing Mean Absolute Error (MAE), Mean Absolute Percentage Error (MAPE), and Mean Square Error (MSE) for evaluation (lower MAE, MAPE and MSE indicate the higher performance). The results are shown in Table 5.

*Table 5.* Evaluation on Travel Time Estimation Task

| | Beijing | | | Porto | | |
|---|---|---|---|---|---|---|
| Methods | MAE | MSE | MAPE | MAE | MSE | MAPE |
| Trembr | 6.66722 | 81.36270 | 90.07389 | 2.12186 | 6.93319 | 31.15763 |
| PIM | 6.98318 | 81.43258 | 92.36455 | 2.09192 | 6.82120 | 27.67270 |
| Toast | 6.84877 | 102.61526 | 62.37184 | 2.18245 | 7.46382 | 29.16603 |
| START | 5.50282 | 70.23768 | 39.74935 | 0.37485 | 0.15862 | 4.43707 |
| LightPath | 4.42273 | 44.37277 | 41.17115 | 0.68104 | 0.73718 | 7.56974 |
| JGRM | 6.91029 | 80.54387 | 90.42306 | 2.13383 | 7.15485 | 26.85475 |
| GTR | **4.01277** | **40.55894** | **33.15476** | **0.01512** | **0.00060** | 0.23882 |
| GTR* | 4.30101 | 49.55222 | 34.32996 | 0.01947 | 0.00093 | **0.19186** |

**Trajectory Classification:** we compare Trembr, START, PIM, Toast, LightPath, JGRM, with our GTR and GTR* for this task, by computing Accuracy (ACC), F1-Score (F1), and Area Under the ROC Curve (AUC) for performance

*Table 6.* The Ablation Study in Porto Dataset

| | Top-$k$ Similarity Search | | | Travel Time Estimation | | | Trajectory Imputation | | | Trajectory Classification | | | Trajectory Generation | | Simplification |
|---|---|---|---|---|---|---|---|---|---|---|---|---|---|---|---|
| | MR | HR@1 | HR@5 | MAE | MSE | MAPE | recall@3 | recall@5 | MAP | ACC | F1 | AUC | Hausdorff | DTW | PED |
| GTR | **1.0028** | **0.9974** | **0.9999** | **0.01512** | **0.00060** | **0.23882** | **0.86465** | **0.93331** | **0.60344** | **0.83213** | **0.83213** | **0.90528** | **0.00286** | **0.01064** | **0.000058** |
| w/o Time Embed | 1.0302 | 0.9872 | 0.9982 | 0.02828 | 0.00177 | 0.31838 | 0.80081 | 0.89473 | 0.50160 | 0.82472 | 0.82471 | 0.90201 | 0.00378 | 0.01590 | 0.000064 |
| w/o Grid Embed | 1.0162 | 0.9978 | 0.9998 | 0.06874 | 0.00716 | 0.71140 | 0.65271 | 0.76521 | 0.37667 | 0.78098 | 0.78099 | 0.88269 | 0.00772 | 0.04193 | 0.000079 |
| w/o Road Embed | 1.0128 | 0.9982 | 0.9998 | 0.03810 | 0.00228 | 0.50857 | 0.78029 | 0.87494 | 0.48701 | 0.81981 | 0.81981 | 0.90345 | 0.00472 | 0.02622 | 0.000061 |
| w/o ST-MOE | 1.0438 | 0.9790 | 0.9990 | 0.08088 | 0.00943 | 0.89989 | 0.71479 | 0.82876 | 0.41398 | 0.81616 | 0.81616 | 0.90054 | 0.00777 | 0.03067 | 0.000081 |
| w/o TripletLoss | 1.0162 | 0.9880 | 0.9998 | 0.01837 | 0.00081 | 0.24431 | 0.86068 | 0.93111 | 0.59674 | 0.82882 | 0.82881 | 0.90462 | 0.00293 | 0.01068 | 0.000062 |
| w/o MLMLoss | 741.1848 | 0.0326 | 0.0670 | 0.01752 | 0.00147 | 0.36225 | 0.82449 | 0.90508 | 0.55668 | 0.81761 | 0.81762 | 0.90313 | 0.00304 | 0.01219 | 0.000067 |

evaluation. Specifically, higher ACC, F1-Score, and AUC indicate higher classification accuracy. The results are depicted in Table 7.

*Table 7.* Evaluation on Trajectory Classification Task

| | Beijing | | | Porto | | |
|---|---|---|---|---|---|---|
| Methods | ACC | F1 | AUC | ACC | F1 | AUC |
| Trembr | 0.80132 | 0.85030 | 0.85937 | 0.80846 | 0.80845 | 0.88425 |
| PIM | 0.68116 | 0.68111 | 0.65818 | 0.73370 | 0.73371 | 0.85097 |
| Toast | 0.68114 | 0.81031 | 0.50000 | 0.50379 | 0.50376 | 0.50100 |
| START | 0.73961 | 0.81045 | 0.79378 | 0.78320 | 0.78321 | 0.86379 |
| LightPath | 0.74454 | 0.82524 | 0.79614 | 0.74303 | 0.74303 | 0.86241 |
| JGRM | 0.76933 | 0.84583 | 0.83394 | 0.62007 | 0.62006 | 0.78775 |
| GTR | 0.80164 | 0.85509 | 0.86297 | 0.83213 | 0.83213 | 0.90528 |
| GTR* | **0.80185** | **0.85535** | **0.86632** | **0.83904** | **0.83904** | **0.91317** |

**Trajectory Generation:** we evaluate the similarities between the generated and original trajectories, by using distance measures Hausdorff (Xie et al., 2017) and DTW (Keogh & Ratanamahatana, 2005). Note that, the closer the distance between the generated and original trajectories, the more effective the trajectory generation method. We compare our GTR and GTR* with SeqGAN, ControlTraj, and TS-TrajGen, where the results are shown in Table 8.

*Table 8.* Evaluation on Trajectory Generation Task

| | Beijing | | Porto | |
|---|---|---|---|---|
| Methods | Hausdorff | DTW | Hausdorff | DTW |
| SeqGAN | 0.06527 | 0.59283 | 0.02752 | 0.40514 |
| ControlTraj | 0.03139 | 0.51857 | 0.00608 | 0.03579 |
| TS-TrajGen | 0.04861 | 0.54997 | 0.01109 | 0.21610 |
| GTR | 0.03080 | 0.48664 | 0.00286 | 0.01064 |
| GTR* | **0.03047** | **0.48578** | **0.00262** | **0.01011** |

**Overall, we have the following observations.**

(i) Our GTR effectively generates general trajectory representations that support a wide range of downstream tasks, in contrast to state-of-the-art models, which are often tailored to specific tasks. This versatility is achieved through pre-training based on comprehensive feature extraction (cf. Section 3.1), enabling the model to meet the diverse and complex requirements of various tasks.

(ii) GTR consistently outperforms state-of-the-art models across all tasks, achieving accuracy improvements: up to 15%–60% for trajectory imputation, 1%–4% for trajectory classification, 10%–90% for the TTE task, 6%–26% for trajectory simplification, 4%–8% for trajectory similarity computation, and 37%–81% for trajectory generation. These gains are attributed to our use of the MVE and STP modules, which dynamically integrate spatio-temporal features in a multi-view setting. This design enables the trajectory representation learning model to dynamically capture and preserve relationships among trajectories in the learned representations, thereby enhancing performance across diverse trajectory analysis tasks.

(iii) GTR* significantly enhances the performance of GTR across most trajectory analysis tasks by incrementally updating model parameters online. This effectively leverages both historical and newly available trajectory data, allowing continuous optimization and achieving superior performance.

### 4.3. Ablation Study (*RQ3*)

We also conducted ablation studies to prove the effectiveness of six key components within our GTR on six mainstream tasks. **(1) w/o Time Embed:** This variant removes the time embedding. **(2) w/o Grid Embed:** This variant removes the grid embedding, including the grid POIs features. **(3) w/o Road Embed:** Similar to the previous one, which mainly removes the GAT and replaces it with embedding vectors. **(4) w/o ST-MoE:** This variant removes the ST-MoE module. **(5) w/o TripletLoss:** This variant removes the triplet task for pre-training. **(6) w/o MLMLoss:** This variant removes the MLM task for pre-training. Table 6 presents the results in Porto, while the results in Beijing are shown at Table 10. We observe that our GTR framework achieves the best overall performance across most variants. This indicates that the proposed MVE and STP modules are helpful in supporting general trajectory representation learning.

### 4.4. *RQ4 & RQ5*

To answer *RQ4* and *RQ5*, more experiments can refer to the Appendix C.4 and Appendix C.6, respectively.

## 5. Conclusions

In this paper, we propose GTR, a general, multi-view, and dynamic framework for trajectory representation learning. GTR learns spatio-temporal features via a designed multi-view encoder. To adaptively integrate spatial and temporal features, we introduce mixture of experts, tailored for downstream tasks. GTR also incorporates online frozen-hot strategy to dynamic updating. Future work will explore integrating LLMs to broaden GTR's applicability.

## Acknowledgements

This work was supported by the NSFC under Grant No. 62402422 and 62472377, Yongjiang Talent Introduction Programme (2024A-162-G), Zhejiang Provincial Natural Science Foundation of China under Grant No. LZ25F020001, and ZTE Industry-University-Institute Cooperation Funds under Grant No. IA20240731007. Ziquan Fang is the corresponding author.

## Impact Statement

This paper presents work whose goal is to advance the field of Machine Learning. There are many potential societal consequences of our work, none of which we feel must be specifically highlighted here.

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

# Appendix

*Table 9.* A Comparison of the Latest Trajectory Representation Learning Models

| Model | Year | Trajectory Analysis Tasks | | | | | | Multi-View | Model Updating |
|---|---|---|---|---|---|---|---|---|---|
| | | Similarity | TTE | Simplification | Imputation | Generation | Classification | | |
| Trembr (Fu & Lee, 2020) | TIST'20 | ✓ | ✓ | ✗ | ✗ | ✗ | ✓ | ✗ | ✗ |
| PIM (Yang et al., 2021b) | IJCAI'21 | ✓ | ✓ | ✗ | ✗ | ✗ | ✓ | ✗ | ✗ |
| Toast (Chen et al., 2021) | CIKM'21 | ✓ | ✓ | ✗ | ✗ | ✗ | ✓ | ✗ | ✗ |
| JCLRNT (Mao et al., 2022) | CIKM'22 | ✓ | ✓ | ✗ | ✗ | ✗ | ✓ | ✗ | ✗ |
| START (Jiang et al., 2023a) | ICDE'23 | ✓ | ✓ | ✗ | ✗ | ✗ | ✓ | ✗ | ✗ |
| LightPath (Yang et al., 2023) | KDD'23 | ✓ | ✓ | ✗ | ✗ | ✗ | ✓ | ✗ | ✗ |
| MMTEC (Lin et al., 2023) | TKDE'23 | ✓ | ✓ | ✗ | ✗ | ✗ | ✓ | ✓ | ✗ |
| JGRM (Ma et al., 2024) | WWW'24 | ✓ | ✓ | ✗ | ✗ | ✗ | ✓ | ✓ | ✗ |
| GTR | 2025 | ✓ | ✓ | ✓ | ✓ | ✓ | ✓ | ✓ | ✓ |

# A. Related Work

**Trajectory Representation Learning in Free Space.** In free space, the GPS trajectories are transformed into data sequences for representation learning. For instance, traj2vec (Yao et al., 2017) splits trajectories into time interval based windows and encodes each window as a token in a sequence. Similarly, t2vec (Li et al., 2018) and E$^2$DTC (Fang et al., 2021) partition the space into equally sized grids and use seq2seq models for representation. NeuTraj (Yao et al., 2019) introduces an enhanced spatial memory module to capture correlations among trajectories, while T3S (Yang et al., 2021a) incorporates auxiliary loss functions and self-attention mechanisms to improve trajectory representations for the similarity search task. TrajFormer (Liang et al., 2022) performs continuous point embedding to learn the representations of each point. S3 (Fang et al., 2023) designs a lightweight framework with two chained seq2seq models to support the trajectory simplification task. Furthermore, TrajCL (Chang et al., 2023a) applies contrastive learning based on data augmentation and dual attention for trajectory similarity tasks. However, all of these studies ignore the physical constraints imposed by road networks upon behaviors of mobile road users, e.g., people and vehicles. In this paper, we target performing robust trajectory representation learning considering road network context.

**Trajectory Representation Learning in Road Networks.** In road networks, the original GPS trajectories are typically mapped onto a road network using map-matching algorithms (Yang & Gidofalvi, 2018). Compared to free-space settings, it provides the topological structure of the road network, enabling more precise modeling of trajectories. In this setting, existing works are primarily divided into two classes, **task-specific** and **general-purpose** methods. For the former, GTS (Han et al., 2021) designs a GNN-based framework for similarity computation tasks. ST2vec (Fang et al., 2022) considers temporal trajectory similarity and fuses spatial and temporal features. Trajbert (Si et al., 2023) devises a BERT-based (Devlin et al., 2019) trajectory recovery method with a spatial-temporal aware loss function. Furthermore, TS-TrajGen (Jiang et al., 2023b) introduces a two-stage generative adversarial framework to support trajectory generation tasks. To support more tasks, recently, there have been several general-purpose models. Trembr (Fu & Lee, 2020) extends t2vec by performing map matching and introducing a road2vec module to learn road network representations. TrajGAT (Yao et al., 2022) integrates GATs with Transformer to learn trajectory embedding, which retains long-term dependencies. Toast (Chen et al., 2021) and PIM (Yang et al., 2021b) utilize node2vec to learn road representations and then apply a Transformer encoder for trajectory embeddings. Further methods, including JCLRNT (Mao et al., 2022) and SARN (Chang et al., 2023b) enhance the Toast model through contrastive learning. START (Jiang et al., 2023a) incorporates semantic information from road networks and combines GAT with BERT for trajectory representations tailored for different tasks. MMTEC (Lin et al., 2023) utilizes discrete and continuous encoders to learn a general representation. LightPath (Yang et al., 2023) employs a relational inference contrastive approach with a global knowledge distillation framework for encoding. JGRM (Ma et al., 2024) uses a Transformer to learn representations from continuous GPS points and the road network. Table 9 summarizes existing general-purpose trajectory representation learning methods. As observed, although MMTEC (Lin et al., 2023) and JGRM (Ma et al., 2024) integrate multiple views, they still overlook the hidden POI features within different regions, thereby affecting the performance of downstream tasks. Moreover, existing methods support limited trajectory analysis tasks while failing to enable representation model updates. In this paper, we propose a multi-view trajectory representation framework that jointly captures free-space semantics and road-network topology features. Furthermore, our approach supports the widest range of trajectory analysis tasks and enables online model updates, addressing the limitations of prior methods.

# B. Additional Methodology Details

### B.1. Fine-tuning Methods

As illustrated in Fig. 6, a simplified workflow for each task is provided. All of the fine-tuning methods are listed as follows.

*Table 10.* The Ablation Study in Beijing Dataset

| | Top-k Similar Trajectory Query | | | Travel Time Estimation | | | Trajectory Imputation | | | Trajectory Classification | | | Trajectory Generation | | Simplification |
|---|---|---|---|---|---|---|---|---|---|---|---|---|---|---|---|
| | MR | HR@1 | HR@5 | MAE | MSE | MAPE | recall@3 | recall@5 | MAP | ACC | F1 | AUC | Hausdorff | DTW | PED |
| GTR | **1.0130** | **0.9906** | **0.9996** | **4.01277** | **40.55894** | 33.15476 | **0.99406** | **0.99521** | **0.98652** | **0.80164** | **0.85509** | **0.86297** | **0.03080** | **0.48664** | **0.000035** |
| w/o Time Embed | 1.0406 | 0.9750 | 0.9986 | 4.09158 | 40.70333 | 32.69300 | 0.99401 | 0.99507 | 0.98601 | 0.77005 | 0.83011 | 0.83322 | 0.03089 | 0.50565 | 0.000035 |
| w/o Grid Embed | 1.0274 | 0.9810 | 0.9992 | 4.08252 | 40.80434 | 31.53830 | 0.98674 | 0.98987 | 0.96646 | 0.76626 | 0.82965 | 0.82825 | 0.03739 | 0.52296 | 0.000044 |
| w/o Road Embed | 1.0162 | 0.9880 | 0.9995 | 4.02020 | 40.59894 | **30.52987** | 0.98293 | 0.98685 | 0.95918 | 0.75664 | 0.81914 | 0.81847 | 0.03963 | 0.53026 | 0.000050 |
| w/o ST-MOE | 1.0174 | 0.9882 | 0.9994 | 4.05528 | 40.86599 | 30.82057 | 0.98327 | 0.98772 | 0.95649 | 0.74789 | 0.80434 | 0.82289 | 0.11627 | 0.95925 | 0.000049 |
| w/o TripletLoss | 1.0274 | 0.9798 | 0.9994 | 4.14899 | 41.13403 | 32.97659 | 0.99318 | 0.99467 | 0.98353 | 0.79440 | 0.85313 | 0.85359 | 0.03087 | 0.50513 | 0.000036 |
| w/o MLMLoss | 739.6854 | 0.0786 | 0.1122 | 4.18216 | 41.26676 | 36.63815 | 0.98352 | 0.98882 | 0.95294 | 0.76601 | 0.83881 | 0.82127 | 0.03097 | 0.48787 | 0.000041 |

(a) Similarity     (b) Simplification     (c) Imputation     (d) TTE     (e) Classification     (f) Generation

*Figure 6.* The Mainstream Trajectory Analysis Tasks

1. **Trajectory Similarity Computation.** Fine-tuning is unnecessary for the similarity task since the model learns trajectory distinctions during the pre-training stage. Instead, we evaluate model performance using the most similar trajectory search and visualize the Top-$k$ similar trajectory search results for comparison with other models.

2. **Trajectory Simplification.** For the trajectory simplification task, we first generate labels using the Douglas-Peucker simplification algorithm (Douglas & Peucker, 1973). Then, we employ a binary classification task to determine whether specific segments of the trajectory should be omitted:

$$\mathcal{L}_{simplify} = -\frac{1}{N}\sum_{i=1}^{N}\left(y_i\log(\hat{y}_i) + (1-y_i)\log(1-\hat{y}_i)\right), \tag{21}$$

where $N$ is the number of training samples, $\hat{y}_i$ denotes the predicted probability value, and $y_i$ represents the true label.

3. **Trajectory Imputation.** For the trajectory imputation task, we replace $20\%$ of the trajectory with masked tokens, similar to the pre-training task, and then predict the missing segments using the model. The original trajectory serves as the label, with the cross-entropy loss function used as the optimization objective:

$$\mathcal{L}_{imputation} = \frac{1}{N_{mask}}\sum_{i=1}^{N_{mask}}\sum_{c_v=1}^{C_v} -y_i(c_v)\log(\hat{y}_i(c_v)), \tag{22}$$

where $C_v$ is the size of the trajectory vocabulary, $N_{mask}$ is the number of masked tokens, $\hat{y}_i$ denotes the predicted probability value of the masked token, and $y_i$ represents the true label.

4. **Trajectory Classification.** This task aims to classify trajectories based on specific labels, such as whether they are carrying passengers or the type of taxi call. We utilize a simple fully connected layer followed by a *Softmax* activation to obtain the predictions, expressed as $\hat{y} = Softmax(FC(R_r))$. The model is then optimized using the cross-entropy loss:

$$\mathcal{L}_{classification} = \frac{1}{N}\sum_{i=1}^{N}\sum_{c_v=1}^{C_v} -y_i(c_v)\log(\hat{y}_i(c_v)), \tag{23}$$

where $C_v$ is the size of the trajectory vocabulary, $N$ is the number of training samples, $\hat{y}_i$ denotes the predicted probability value by the model, and $y_i$ represents the true label.

5. **Trajectory Generation.** The trajectory generation task involves removing 50% of the trajectory's content and then predicting the remaining 50% using the model. The predicted results are evaluated using the cross-entropy loss, and the outcome is compared with the original trajectory using common trajectory metrics, such as Dynamic Time Warping (DTW), to assess the effectiveness of the generation:

$$\mathcal{L}_{generation} = \frac{1}{N_{mask}}\sum_{i=1}^{N_{mask}}\sum_{c_v=1}^{C_v} -y_i(c_v)\log(\hat{y}_i(c_v)), \tag{24}$$

where $C_v$ is the size of the trajectory vocabulary, $N_{mask}$ is the number of masked tokens, $\hat{y}_i$ denotes the predicted probability value of the masked token, and $y_i$ represents the true label.

---

**Algorithm 1** The Process of GTR

---

**Input:** road network $G = (V, E, A)$, GPS trajectory $\mathcal{T}$, road features $F_v$, grids types $c^{poi}$
1: **Preprocess:** map $\mathcal{T}$ on $G$ and space to get $\mathcal{T}^r$ and $\mathcal{T}^g$, pre-training dataset $\mathcal{D}^P$, fine-tuning dataset $\mathcal{D}^F$, updating dataset $\mathcal{D}^U$
2: **for** $d_{mask}, d_a, d_p, d_n \in \mathcal{D}^P$ **do**
3:     Calculate $\mathcal{L} = \gamma \cdot \mathcal{L}_{triplet}(d_a, d_p, d_n) + (1 - \gamma) \cdot \mathcal{L}_{mask}(d_{mask})$;
4:     Update GTR by minimizing $\mathcal{L}$;
5: **end for**
6: **for** $d_{task} \in \mathcal{D}^F$ **do**
7:     Calculate $\mathcal{L}_{task}$ for the downstream tasks;
8:     Update GTR by minimizing $\mathcal{L}_{task}$;
9: **end for**
10: **Online updating stage:** Freeze the half of GTR's transformer encoder layers;
11: **for** $d_{update} \in \mathcal{D}^U$ **do**
12:     Calculate $\mathcal{L}_{update}$ for the downstream tasks;
13:     Update GTR by minimizing $\mathcal{L}_{update}$;
14: **end for**

---

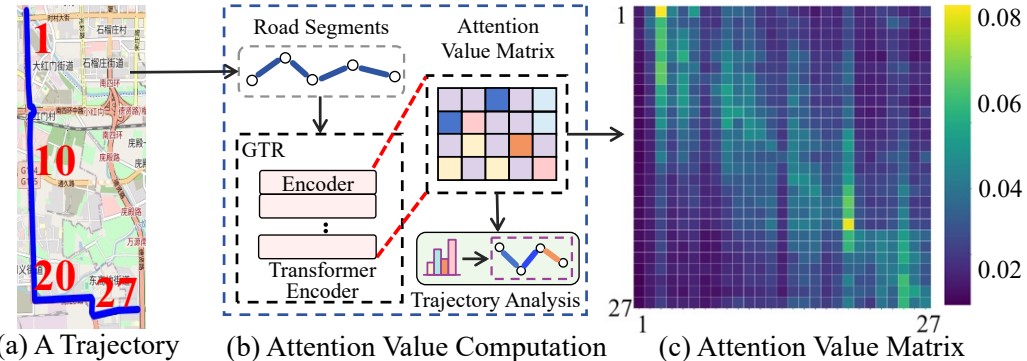

(a) A Trajectory     (b) Attention Value Computation     (c) Attention Value Matrix

*Figure 7.* Visualization of Attention Values

## B.2. The Process of GTR

Algorithm 1 presents the complete process of GTR training, which consists of preprocessing, pre-training, fine-tuning, and online updating.

In the preprocessing stage (line 1), we require the GPS trajectory $\mathcal{T}$, the road network $G = (V, E, A)$ along with its static road features $F_v$ and POI types $c^{poi}$ for each grid. We then apply the map-matching algorithm and grid partition method to transform $\mathcal{T}$ into $\mathcal{T}^r$ and $\mathcal{T}^g$. The features $F_v$ are used as the initial embedding for the GAT in GTR's MVE module. Next, we process $\mathcal{T}^r$, $\mathcal{T}^g$, and $c^{poi}$ to generate the pre-training dataset $\mathcal{D}^P$, fine-tuning dataset $\mathcal{D}^F$, and updating dataset $\mathcal{D}^U$.

In the pre-training stage (lines 2–5), we first obtain the pre-training data $d_{mask}, d_a, d_p, d_n$ from $\mathcal{D}^T$. We then compute the total loss by combining the triplet loss $\mathcal{L}_{triplet}$ and MLM loss $\mathcal{L}_{mask}$ with the weight parameter $\gamma$ to update GTR. In the fine-tuning stage(lines 6–9), we fine-tune GTR by calculating the downstream task loss $\mathcal{L}_{task}$ for different downstream tasks, as outlined in Section 3.2.3.

Finally, in the online updating stage (lines 10–14), we use the fine-tuned GTR from the previous stage, freeze half of its Transformer encoder layers, and update the GTR by minimizing the updating loss $\mathcal{L}_{update}$.

## B.3. Model Interpreting

Understanding how different parameters in the model influence the representation vectors is crucial. Therefore, an interpretable model evaluation method is necessary for optimization. A natural approach is to analyze the various model parameters in detail. However, some parameters contribute little to model interpretation, while consuming a significant amount of time during the analysis.

We propose a simple yet effective method to address this gap. As shown in Fig. 7(a), the trajectory constrained by the road network contains 27 road segments. These segments are processed through our GTR framework, illustrated in Fig. 7 (b), where they pass through the Transformer encoder layers. From the encoder, we extract the attention value matrix, which represents the pairwise attention weights among the 27 road segments. This matrix indicates the degree of attention one

road segment pays to another, enabling the model to effectively encode contextual information by learning the relationships between different segments of the trajectory. Fig. 7(c) visualizes the attention value matrix as a heat map, reflecting the influence of neighboring road segments on the trajectory in Fig. 7(a). The higher the attention value, the greater the impact of a road segment. Based on these attention values, we perform an explanatory analysis, comparing the attention values derived from the Transformer encoder (cf. Eq. 2), which provides actionable insights for model optimization. Importantly, no additional computational or time costs are incurred during model interpretation, as we utilize information generated during model training.

### B.4. Complexity Analysis

For the process of GTR, the time complexity of the single training phase is $O(4 \cdot |\mathcal{D}^P| + |\mathcal{D}^F| + |\mathcal{D}^U|)$, depending on the size of the datasets. In the pre-training stage, we calculate the $\mathcal{L}_{triplet}$ and $\mathcal{L}_{mask}$ require a complexity of $O(4 \cdot |\mathcal{D}^P|)$. While for the fine-tuning stage, we calculate $\mathcal{L}_{task}$ for different downstream tasks, having a complexity of $O(|\mathcal{D}^F|)$ for this stage. Finally, in the online updating stage, the complexity of calculating and minimizing $\mathcal{L}_{update}$ is $O(|\mathcal{D}^U|)$.

### B.5. Data Enhance Strategies

To enable the model to learn from more diverse data for enhancing the effectiveness of pre-training, we employed the following data augmentation strategies:

**1) Sub-trajectory Selecting:** This augmentation enhances trajectories by randomly removing a continuous subsequence. To maintain the trajectory's continuity, trimming is applied only at the start or end of the trajectory. The trimming ratio is randomly selected between 0.05 and 0.15. This method is effective because trajectories with similar starting points or destinations often share similar features.

**2) Road Drift:** In tasks involving road drift, random roads and their corresponding timestamps within a trajectory are selected and masked. The resulting masked trajectories, which are treated as having missing values, enable the model to learn travel semantics across both temporal and spatial dimensions.

### B.6. POIs Extraction Approach

We propose a POIs extraction approach. First, we obtain the POIs from OpenStreetMap and classify the different types of POIs into mainly 4 categories: service POIs, residential POIs, commercial POIs, and other POIs. Then we calculate the number of each POI type within every grid. As the result of the residential areas always containing commercial POIs like small restaurants or shops, we also measure the size of each area to accurately determine the type of each grid.

## C. Additional Experimental Details

### C.1. Extra Experiment Settings

In the pretraining stage, we set the hidden size to 768, the number of training epochs to 10, and both the attention layers and heads to 12. In the fine-tuning stage, we set the training epochs to 50. For the most similar trajectory search, the query dataset consists of $5k$ trajectories and the key dataset consists of $50k$ trajectories, with a detour rate of 0.2. In the classification task, there are two labels in the Beijing dataset and three labels in the Porto dataset. In the travel time estimation task, we predict the trip duration in minutes.

### C.2. Model Complexity Analysis

A comparison of model parameters is shown in Table 11. While GTR has a higher parameter count due to its multi-view encoder (MVE) and spatio-temporal fusion pre-training (STP) modules, this increase is justified by two key advantages. (i) **Enhanced Capability**. The additional parameters enable GTR to support more downstream tasks effectively. (ii) **Performance Gains**. The trade-off in model size is offset by improvements in accuracy and robustness.

### C.3. Model Efficiency Study

In this part, we focus on testing the efficiency of our approach. We measure the training time of the pre-training stage and the inference time of the most similar trajectory search. We choose $5k$ query trajectories and $50k$ key trajectories from the

| Model Name | Parameter Size (MB) |
|---|---|
| PIM | 94.57 |
| Trembr | 148.08 |
| Toast | 161.18 |
| START | 1126.40 |
| LightPath | **73.96** |
| JGRM | 375.60 |
| GTR | 862.99 |

*Table 11.* Comparison of model parameter sizes.

| | Training Time (minutes) | Inference Time (minutes) |
|---|---|---|
| Trembr | 7.450 | **1.450** |
| PIM | **5.867** | 2.873 |
| Toast | 9.883 | 1.886 |
| START | 51.617 | 4.850 |
| LightPath | 7.717 | 4.733 |
| JGRM | 118.598 | 3.617 |
| GTR | 179.583 | 6.883 |

*Table 12.* Comparison of Training and Inference Time Across Models.

test dataset. The valuation of the efficiency result is shown in Table 12. GTR spends more training time than other models but does not spend too much time during inference and outperforms state-of-the-art models in all tasks.

### C.4. Model Interpretability Evaluation (*RQ4*)

We consider explaining the model training procedure for two tasks (i.e., travel time estimation and trajectory classification). Because they are the typical tasks that require both spatial and temporal features, and enable effective evaluation of our model's interpreting strategy. Thus, in this subsection, we conduct model interpretability evaluation for the two tasks above. Specifically, we select the 5 important road segments $V_{imp}$ with the greatest attention values, and mask them by replacing or removing. Then, we evaluate GTR by using these processed trajectories (i.e., GTR w/o $V_{imp}$) for the two tasks, and compare it with GTR trained by using the original trajectories (i.e., GTR w/ $V_{imp}$).

As shown in Figure 8, we observe that GTR w/ $V_{imp}$ performs better than GTR w/o $V_{imp}$ for both travel time estimation and trajectory classification tasks on two datasets. This is because the important segments play a vital role in trajectory representation learning. Thus, it is appropriate to assign more neurons or network layers for the road segments with higher attention values, instead of embedding all of the road segments uniformly. This provides optimization guidelines for improving model structure, achieving more effective trajectory representations.

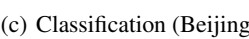

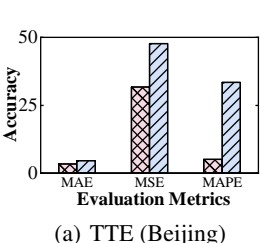 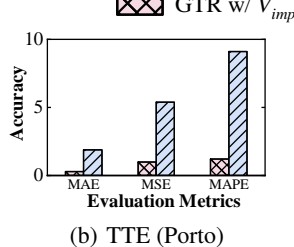 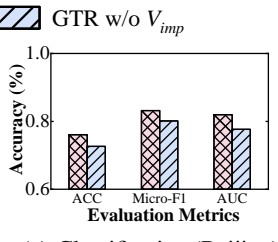 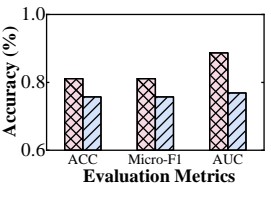

(a) TTE (Beijing)    (b) TTE (Porto)    (c) Classification (Beijing)    (d) Classification (Porto)

*Figure 8.* Interpretability Evaluation

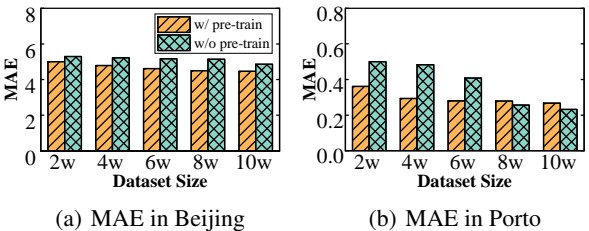

(a) MAE in Beijing      (b) MAE in Porto

*Figure 9.* Pre-training Effect Study

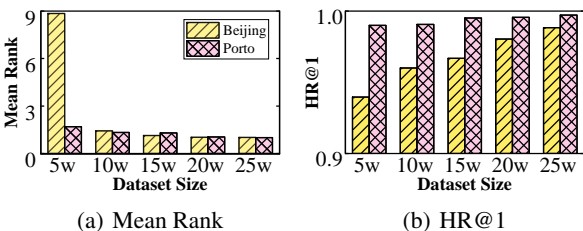

(a) Mean Rank      (b) HR@1

*Figure 10.* Model Scalability Evaluation

### C.5. Extra Experiment on Chengdu Dataset

Existing works (START (Jiang et al., 2023a), Trembr (Fu & Lee, 2020), ST2Vec (Fang et al., 2022), etc.) mainly use Beijing and Porto datasets, so we adopted them for fair comparison. To test the robustness of GTR, we have added the larger Chengdu dataset (containing 2,140,129 trajectories). We specifically test the most computationally intensive trajectory similarity computation task. Results are shown in Table 13. As expected, GTR maintains superior performance over baselines on the Chengdu dataset, confirming its robustness.

*Table 13.* Evaluation on Top-$k$ Similarity Computation Task

| Methods | Chengdu | | |
|---|---|---|---|
| | Mean Rank | HR@1 | HR@5 |
| Trembr | 63.5602 | 0.1940 | 0.4060 |
| PIM | 7.1468 | 0.6724 | 0.8552 |
| Toast | 8.3456 | 0.5588 | 0.7874 |
| START | 6.8745 | 0.6575 | 0.8434 |
| LightPath | 6.2140 | 0.6016 | 0.8206 |
| JGRM | 2.3111 | 0.8292 | 0.9352 |
| GTR | **1.7401** | **0.9212** | **0.9834** |

### C.6. Model Scalability Evaluation (*RQ5*)

We conduct the model capacity evaluation by performing the most similar search using GTR trained on varying dataset sizes. The results are reported in Figure 10. We observe that with the growth of dataset size, our model performs better (i.e., MR<2 and HR@1>0.95) on both the Beijing and Porto datasets. This is because GTR is able to extract more spatial and temporal features via multi-view encoding from more training samples. Therefore, GTR has a large capacity to support large-scale model training and data processing.

### C.7. Case Study of the Top-3 Similarity Search

In this part, we present a case study for comparing the top three performing models on Top-3 similar trajectory search. We randomly select one trajectory from the test dataset, and find the Top-3 similar trajectories within it. The results are shown in Figure 11. We observe that our GTR can find more similar trajectories than other models due to the effective MVE module and STP module in GTR.

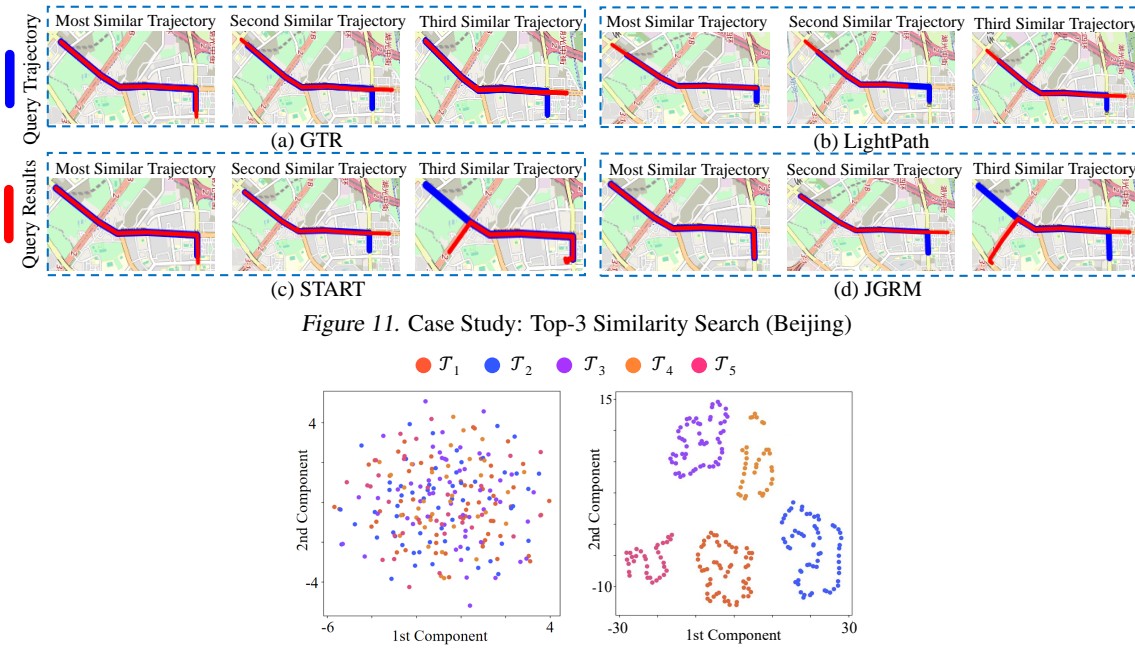

*Figure 11.* Case Study: Top-3 Similarity Search (Beijing)

(a) w/o pre-train    (b) w/ pre-train

*Figure 12.* Case Study of Trajectory Representations in Beijing

## C.8. Case Study of Trajectory Representation

In this experiment, we study the effect of the pre-training model within GTR in two ways. First, we train the GTR model without pre-training (i.e., w/o pre-train), and compare it with the pre-trained model (i.e., w/ pre-train) on the travel time estimation task. The results are reported in Figure 9. Second, we conduct a case study, which visualizes the trajectory representations generated by GTR trained with and without pre-training, respectively. Figure 12 presents the visualizations. We observe that w/ pre-train performs better than w/o pre-train on various dataset sizes of the two datasets, indicating that the pre-trained model is vital in GTR and able to capture more general knowledge for serving downstream tasks.

