# OpenReview forum: "GTR: A General, Multi-View, and Dynamic Framework for Trajectory Representation Learning"
_ICML.cc/2025/Conference — ICML 2025 poster_

### Official Review · Reviewer_nfbU · 2025-03-08

**Overall Recommendation:** 4

**Summary:**

This paper proposes GTR, a general, multi-view, dynamic framework for learning trajectory representation. The authors conduct a thorough review of existing studies and identify three critical limitations in current research: (1) reliance on single-view representations, (2) limited multitasking capabilities, and (3) insufficient support for model updates. To address these challenges, GTR proposes three innovative components: a multi-view encoder (MVE) for capturing diverse perspectives of trajectory data, a spatio-temporal fusion pre-training (STP) mechanism for enhanced multitasking performance, and an online frozen-hot updating (OFU) strategy to facilitate dynamic model updates. Extensive experimental evaluations demonstrate that GTR consistently surpasses 12 state-of-the-art methods across six mainstream trajectory analysis tasks. Furthermore, the experiments validate the superiority and effectiveness of the proposed module designs, as well as the model's scalability and efficiency. These results highlight GTR's potential as a robust and versatile solution for trajectory representation learning.

**Claims And Evidence:**

The motivation behind this study is clearly articulated, and the experimental results validate the identified shortcomings of existing work.

**Essential References Not Discussed:**

The literature review is sufficient and the SOTA works have been discussed and evaluated.

**Experimental Designs Or Analyses:**

This paper conducts a comprehensive and rigorous set of experiments, thoroughly evaluating the proposed GTR framework from multiple perspectives, including model effectiveness, ablation, efficiency, scalability, case studies, etc.

**Methods And Evaluation Criteria:**

The studied trajectory representation learning problem is important in spatio-temporal data mining community. The authors have selected datasets and tasks that are widely recognized and extensively utilized in the field.

**Other Comments Or Suggestions:**

None

**Other Strengths And Weaknesses:**

Strengths:

S1. The studied trajectory representation learning problem holds significant importance within the spatio-temporal data mining community, as it addresses fundamental challenges in analyzing and understanding complex movement patterns, which are critical for various real-world applications.

S2. The paper has effectively identified three key limitations in previous studies. Then, the paper proposes three innovative solutions through the multi-view encoder (MVE), spatio-temporal fusion pre-training (STP), and online frozen-hot updating (OFU) mechanisms.

S3. The extensive experiments have convincingly demonstrated the superior performance of GTR and its individual modules over state-of-the-art baselines across multiple tasks and datasets.

S4. The paper is well-structured and easy to follow, making it accessible to readers while effectively conveying its technical contributions and experimental results.

Weaknesses:
W1. Some parts of this paper are not explained very clearly, which could lead to misunderstandings and ambiguities. i) First, since there are many symbols, a symbol table is recommended for better following. ii) Besides, although the appendix includes detailed descriptions of relevant studies, the absence of citations of this "Section" in the main body may create the impression that the paper lacks a thorough discussion of prior studies. iii) In Table 1, does "Avg.Length" refer to the spatial length of the trajectory or the number of trajectory points?

W2. The experimental section can be optimized. i) For instance, including the percentage increase in experimental tables can indeed enhance the clarity and impact of the results. ii) The calculation processes of some evaluation metrics (e.g., PED for trajectory simplification task, Hausdorff/DTW for trajectory generation task, HR@x for trajectory similarity search task) are not provided.

W3. There are also some minor details that need attention. i) "we conduct extensive experiments on two real-world datasets demonstrate"->"we conduct extensive experiments on two real-world datasets to demonstrate". ii) "In this paper, we target perform"->"In this paper, we target performing". iii) "In classification task, there are two labels in beijing dataset"->"In classification task, there are two labels in the Beijing dataset". iv) "Mean Absolut Percentage Error (MAPE)"->"Mean Absolute Percentage Error (MAPE)"

**Questions For Authors:**

Please response to the comments in W1-W3.

**Relation To Broader Scientific Literature:**

The paper has identified three key limitations in previous studies—single-view representation, limited multitasking capabilities, and lack of support for model updates—and proposes innovative solutions through the multi-view encoder (MVE), spatio-temporal fusion pre-training (STP), and online frozen-hot updating (OFU) mechanisms. An extensive evaluation using various tasks and datasets show the superiorities of the framework and its designs.

**Theoretical Claims:**

No mathematical proofs are provided in the paper.

---

> ### Author Rebuttal · Authors · 2025-04-01
>
> **We appreciate the positive comments and our responses are detailed below.**
>
> ```
> W1: Some parts of this paper are not explained very clearly, which could lead to misunderstandings and ambiguities. i) First, since there are many symbols, a symbol table is recommended for better following. ii) Besides, although the appendix includes detailed descriptions of relevant studies, the absence of citations of this "Section" in the main body may create the impression that the paper lacks a thorough discussion of prior studies. iii) In Table 1, does "Avg.Length" refer to the spatial length of the trajectory or the number of trajectory points?
> ```
> We sincerely appreciate the reviewer’s valuable feedback regarding the clarity of our manuscript. We have addressed each point as follows. (i) We would like to include a notation table below to clearly define all mathematical symbols used throughout the paper. (ii) We will add explicit citations to the appendix’s related work section in the main text. (iii) The “Avg.Length” in Table 1 indeed refers to the average number of trajectory points.
>
> | Symbol | Discription |
> | ----------------- | ------------------------------------- |
> | $\mathcal{T}$     | GPS Trajectory                  |
> | $\mathcal{T^g}$   | Grid Constrained Trajectory           |
> | $\mathcal{T^r}$   | Road-network Constrained Trajectory   |
> | $\mathcal{G}$     | Grid Cells                            |
> | $G$               | Road Network                          |
> | $D^{\mathcal{T}}$ | Road Trajectory Dataset               |
> | $Z_R$             | Road Representation                   |
> | $Z_G$             | Grid Representation                   |
> | $Z_P$             | Position Representation               |
> | $Z_T$             | Temporal Representation               |
> | $Z_S$             | Spatial Representation                |
> | $h$               | Trajectory Generalized Representation |
>
>
> ```
> W2: The experimental section can be optimized. i) For instance, including the percentage increase in experimental tables can indeed enhance the clarity and impact of the results. ii) The calculation processes of some evaluation metrics (e.g., PED for trajectory simplification task, Hausdorff/DTW for trajectory generation task, HR@x for trajectory similarity search task) are not provided.
> ```
> Thanks for the suggestion. (i) In addition to the detailed performance values, we are happy to include the percentage increase, i.e., 15%–60% for trajectory imputation task, 1%–4% for trajectory classification task, 10%–90% for the travel time estimation (TTE) task, 5%–26% for trajectory simplification task, 4%–8% for trajectory similarity computation task, and 37%–81% for trajectory generation task. (ii) As these evaluation metrics are well-established in the field, we directly cited the related papers that introduced them in our manuscript. The definitions of these metrics can refer to the response to reviewer vfe9's.
>
>
> ```
> W3: There are also some minor details that need attention. i) "we conduct extensive experiments on two real-world datasets demonstrate"->"we conduct extensive experiments on two real-world datasets to demonstrate". ii) "In this paper, we target perform"->"In this paper, we target performing". iii) "In classification task, there are two labels in beijing dataset"->"In classification task, there are two labels in the Beijing dataset". iv) "Mean Absolut Percentage Error (MAPE)"->"Mean Absolute Percentage Error (MAPE)"
> ```
> We sincerely appreciate the reviewer's careful reading of our manuscript. We will correct all these minor issues in the revised version. We will carefully proofread the entire manuscript to improve its presentation.

---

> > ### Comment · Reviewer_nfbU · 2025-04-02
> >
> > Thanks for your response. My concerns have been addressed.

---

> > > ### Author Response · Authors · 2025-04-02
> > >
> > > We appreciate your timely response and are glad our responses addressed your concerns. Thanks again for your careful consideration of our work!

---

### Official Review · Reviewer_Th6q · 2025-03-14

**Overall Recommendation:** 4

**Summary:**

This paper introduces GTR, a novel general, multi-view, and dynamic trajectory representation framework. GTR addresses the limitations of conventional approaches that rely exclusively on either free-space or road-network perspectives by incorporating a multi-view encoder to effectively capture the intrinsic spatio-temporal characteristics of trajectory data. GTR further enhances its representation capability through a spatio-temporal mixture of experts mechanism, which dynamically integrates spatial and temporal information. Moreover, GTR proposes an innovative online frozen-hot updating strategy that enables efficient model adaptation. Extensive evaluations conducted on two real datasets, encompassing comparisons with 12 baseline methods, demonstrate that GTR achieves superior performance across multiple metrics, significantly outperforming existing approaches in various trajectory analysis tasks.

## Update after rebuttal: I have  read the rebuttal and my concerns are well addressed. Good luck!

**Claims And Evidence:**

Yes. The authors effectively demonstrate the significance of multi-view, dynamic, and general aspects through comprehensive comparative experiments and a rigorous ablation study.

**Essential References Not Discussed:**

No, much of the work related to this paper has been already discussed.

**Ethical Review Concerns:**

N/A.

**Experimental Designs Or Analyses:**

Yes, I thoroughly examined the soundness and validity of the experimental designs and analyses. The experimental validation in this paper is indeed rigorous and convincing. The study offers a comprehensive performance comparison of 12 baseline models across two real-world datasets. Moreover, ablation studies are systematically conducted to demonstrate the effectiveness of each individual component. In addition, the authors provide details such as efficiency evaluation, model scalability evaluation, etc., which are crucial for demonstrating the practical applicability of the framework in real-world scenarios.

**Methods And Evaluation Criteria:**

Yes. i) To address the generalizability of the framework, this study introduces a novel Spatio-Temporal Mixture of Experts (ST-MOE) module for dynamically learning and integrating spatial and temporal features. ii) To enable online model update, this study proposes an effective online frozen-hot updating strategy.

**Other Comments Or Suggestions:**

D1. In table 5, the TTE task result MAPE in Porto 0.19186 should be bolded.

**Other Strengths And Weaknesses:**

S1. GTR effectively combines free-space and road-network views, providing robust representations for various trajectory tasks, along with an online frozen-hot update mechanism that adapts to the evolving nature of trajectory data.

S2. The paper makes significant experimental contributions, convincingly demonstrating the model's improvements through extensive comparative experiments, comprehensive ablation studies, and efficiency evaluations.

S3. The paper provides an in-depth exploration of the existing research in the trajectory representation learning. It offers a comprehensive discussion of the distinctions between the GTR and prior studies.

S4. The paper is clearly written and easy to follow, with all necessary preliminaries provided. The motivations behind the different components of the model are well articulated.


W1. The paper does not specify the value of the balancing parameter β in the combined loss function (LGTR), leaving ambiguity in how the MLM and triplet tasks are weighted during pre-training, which could affect reproducibility and performance interpretation.

W2. Some figures (e.g., 1 and 2) and tables (e.g. 5 and 8) in the paper are too small to be clearly legible.

**Questions For Authors:**

Please answer the following questions,

Q1. The paper does not specify the value of the balancing parameter β in the combined loss function (LGTR), leaving ambiguity in how the MLM and triplet tasks are weighted during pre-training, which could affect reproducibility and performance interpretation.

Q2. Some figures (e.g., 1 and 2) and tables (e.g. 5 and 8) in the paper are too small to be clearly legible.

**Relation To Broader Scientific Literature:**

This paper makes significant contributions that are closely connected to the wider scientific field. The MVE module addresses the limitations of single-view approaches, while the STP module integrates spatial and temporal features to solve the limitation of task-specific. Furthermore, the OFU module addresses the lack of support for model update. This work addresses the challenges of traditional methods.

**Theoretical Claims:**

Yes.

---

> ### Author Rebuttal · Authors · 2025-04-01
>
> We express our gratitude to the reviewer for providing constructive feedback on our paper, and we greatly appreciate the acknowledgement of our contributions. We have addressed the specific concerns raised by the reviewer as detailed below.
>
> ```
> W1&Q1: The paper does not specify the value of the balancing parameter β in the combined loss function (LGTR), leaving ambiguity in how the MLM and triplet tasks are weighted during pre-training, which could affect reproducibility and performance interpretation.
> ```
> Thanks for pointing this out. In our pre-training framework, **the balancing parameter $\beta$ in the overall loss function is set to $0.7$**, which was determined through empirical validation on the validation set to optimally weigh the MLM and triplet loss components. This configuration ensures a balanced contribution from both tasks while maximizing downstream performance.
>
>
> ```
> W2&Q2: Some figures (e.g., 1 and 2) and tables (e.g. 5 and 8) in the paper are too small to be clearly legible.
> ```
> Thanks for the comment. We will carefully fix these presentation issues in the revised manuscript.
>
>
> ```
> Other Comments Or Suggestions: In table 5, the TTE task result MAPE in Porto 0.19186 should be bolded.
> ```
> Thanks for pointing this out. We will fix this marking error in the revised manuscript.

---

> > ### Comment · Reviewer_Th6q · 2025-04-02
> >
> > Thanks for authors' detailed rebuttal. All my concerns have been well addressed and I lean to vote for acceptance.

---

> > > ### Author Response · Authors · 2025-04-02
> > >
> > > We are happy that our responses have addressed your concerns. We would like to express our sincerest gratitude once again for taking the time to review our paper!

---

### Official Review · Reviewer_z8Rd · 2025-03-20

**Overall Recommendation:** 3

**Summary:**

This paper proposes a novel framework for trajectory representation learning by integrating free-space trajectories with road network-based trajectories. The framework consists of three key components: 1) a multi-view encoder designed to handle different types of trajectories; 2) a spatial-temporal fusion pretraining mechanism that leverages a mixture of spatial and temporal experts; and 3) an online updating strategy for timely processing of new trajectories. Extensive experiments conducted on datasets from Beijing and Porto, comparing against 12 baseline methods, demonstrate the effectiveness of the proposed approach.

**Claims And Evidence:**

Yes

**Essential References Not Discussed:**

1. Xia, Tong, et al. "Attnmove: History enhanced trajectory recovery via attentional network." Proceedings of the AAAI Conference on Artificial Intelligence. Vol. 35. No. 5. 2021.

2. Ren, Huimin, et al. "Mtrajrec: Map-constrained trajectory recovery via seq2seq multi-task learning." Proceedings of the 27th ACM SIGKDD Conference on Knowledge Discovery & Data Mining. 2021.

3. Zhu, Yuanshao, et al. "Controltraj: Controllable trajectory generation with topology-constrained diffusion model." Proceedings of the 30th ACM SIGKDD Conference on Knowledge Discovery and Data Mining. 2024.

**Experimental Designs Or Analyses:**

1. The experiments only evaluate two small GPS trajectory datasets, which restricts the robustness and effectiveness of the proposed methods' performance.

2. While the authors argue that existing works cover only a limited range of trajectory tasks, the tasks mentioned in the paper are also narrow in scope, with notable omissions such as next-location prediction, map matching, and anomaly detection.

3. Most tasks only incorporate baselines based on representation learning, lacking specific SOTA baselines tailored for each task. For example, trajectory imputation tasks fail to include AttnMove and MtrajRec, while trajectory generation tasks overlook ControlTraj and others.

**Methods And Evaluation Criteria:**

The key designs in the proposed framework appear to be trivial, offering limited novelty and technical contribution. The concept of the multi-view encoder is a common and straightforward approach that has been widely adopted in numerous spatial-temporal data mining studies. Additionally, the phrasing of the "mixture of experts" with only one spatial expert and one temporal expert seems somewhat forced and lacks clarity. Furthermore, while the online updating strategy aligns with intuition and represents a simple, practical design, it falls short in terms of technical depth and theoretical grounding.

**Other Comments Or Suggestions:**

NO

**Other Strengths And Weaknesses:**

The paper is well-organized and easy to follow.

**Questions For Authors:**

Please refer to the previous section.

**Relation To Broader Scientific Literature:**

This work is closely related to representation learning in the context of spatial-temporal data mining and trajectory modelling.

**Theoretical Claims:**

NA

---

> ### Author Rebuttal · Authors · 2025-04-01
>
> Thanks for all the valuable comments.
>
> **Response to E1:** Existing works (START, Trembr, ST2Vec, etc.) mainly use Beijing and Porto datasets, so we adopted them for fair comparison. Following the suggestion, we have added the larger Chengdu dataset (containing 2,140,129 trajectories). Due to space limitation, we specifically test the most computationally intensive trajectory similarity computation task. The results are shown in the anonymous link https://anonymous.4open.science/r/Rebuttal_GTR/ (Table 1 : Trajectory Similarity Computation on Chengdu Dataset). As expected, GTR maintains superior performance over baselines on this new dataset, confirming its robustness.
>
> **Response to E2:** **We would like to emphasize that GTR extends task coverage compared to existing methods**. As reported in Table 8 (Appendix A, lines 605--614), GTR is the first unified framework supporting all six fundamental trajectory tasks, whereas prior methods (including the state-of-the-art START, LightPath, JGRM) handle at most three tasks. Note that, our framework allows easy adaptation to additional tasks (e.g., next-location prediction, anomaly detection) through straightforward training strategy modifications. For anomaly detection specifically, our framework can learn discriminative trajectory representations through the MVE and STP modules. Based on that, we can learn representations of different trajectories and classify them as normal or abnormal.
>
> **Response to E3&References:** **We appreciate the reviewer's feedback regarding baseline selection**. We would like to clarify that GTR is fundamentally a trajectory representation learning framework, rather than a task-specific model. Our primary objective is to learn robust trajectory embeddings that can effectively support multiple downstream tasks, which differs from specialized methods designed for individual tasks. This experimental design aligns with existing trajectory representation learning studies. Following the suggestion, we have now included comparisons with task-specific methods (AttnMove, MtrajRec, ControlTraj). The results are shown in the anonymous link https://anonymous.4open.science/r/Rebuttal_GTR/ (Table 2: Evaluationon on Trajectory Imputation Task; Table 3: Evaluationon on Trajectory Generation Task). As observed, GTR still achieves superior performance, further demonstrating the effectiveness. In the revised paper, we will include these results and cite these works.
>
> **Response to Methods And Evaluation Criteria:** We appreciate the concerns about the technical contributions and novelty. GTR effectively and innovatively unified multi-view encoding, multi-task pre-training, and online adaptation, which have been recognized by the other reviewers (vfe9, Th6q, and nfbU).
>
> (i) While multi-view learning is not new, our **MVE is first to integrate Grid, POI, and road-network attributes to jointly model spatial, semantic, and structural features—unlike prior works that focus on only one or two views**. The ablation studies (Tables 4 and 9) prove the effectiveness.
>
> (ii) Our ST-MoE is not a trivial extension of standard MoE, as **we introduces task-aware gating to dynamically adjust spatio-temporal feature fusion**. This enables jointly optimizing multiple tasks while preserving inter-task correlations. Additionally, we employ dedicated spatial and temporal experts (rather than multiple experts) to ensure clear structural separation between spatial and temporal feature learning. This design mitigates the common issue of expert polarization, where only a subset of experts remains active.
>
> (iii) Our online updating strategy is theoretically grounded in incremental learning [1]. We appreciate the opportunity to clarify the theoretical foundations, which **combines layer-wise parameter freezing with Lyapunov stability analysis to ensure robust adaptation while preventing catastrophic forgetting**.
>
> Specifically, we freeze the first $L$ layers and update the last $N−L$ layers. The objective optimization function is:  $\min\_{\theta^{L+1:N}}\mathbb{E}\_{(x, y)}\sim\mathcal{D}\_{\mathrm{new}}[\ell(f\_{\theta\_{\mathrm{pre}}^{1:L}}(x), y;\theta^{L+1:N})]$. With frozen lower-Layer parameters ($(\nabla_{\theta^{1:L}}\ell=0)$), old features remain unchanged, which guarantees that the old features are not forgotten.
>
> The updating process can be modeled as a dynamic system: $\theta_{t+1}^{L+1:N} = \theta_t^{L+1:N} - \eta_t g_t$ , with a Lyapunov function: $V(\theta)=\mathcal{L}\_\mathrm{new}(\theta)+\gamma\|\theta^{1:L}-\theta_\mathrm{pre}^{1:L}\|^2$. As $\gamma\to\infty$, the system satifies: $\mathbb{E}[V(\theta\_{t+1})]\leq\mathbb{E}[V(\theta\_t)]-\eta\_t\|\nabla\mathcal{L}\_{\mathrm{new}}(\theta\_t)\|^2$. The monotonic decrease of $V(\theta)$ ensures stable updates. Freezing lower layers prevents cascading perturbations, balancing new feature learning with old feature retention.
>
> [1] *A Comprehensive Survey of Continual Learning: Theory, Method and Application.*

---

> > ### Comment · Reviewer_z8Rd · 2025-04-02
> >
> > After thorough consideration of the authors' response and fellow reviewers' comments, I acknowledge that the response and revised version have satisfactorily addressed the majority of my initial concerns. In light of these improvements, I have raised my overall score to 3. To further strengthen this valuable contribution, I would like to offer two suggestions for the authors' consideration:
> >
> > 1. Additional refinement of the manuscript's writing and organization to maximize clarity and readability
> >
> > 2. Timely release of comprehensive, well-documented source code to ensure reproducibility and enable broader community engagement with this work
> >
> > These final improvements would elevate the paper's impact and utility for the research community.

---

> > > ### Author Response · Authors · 2025-04-02
> > >
> > > We sincerely appreciate your prompt response and are delighted that our clarifications have addressed your concerns satisfactorily. In the final version of the manuscript, we will carefully incorporate your valuable comments and suggestions to further enhance the writing, organization, and clarity. Additionally, we will ensure that the source code is thoroughly documented to guarantee reproducibility. Thank you once again for your insightful feedback.

---

### Official Review · Reviewer_vfe9 · 2025-03-23

**Overall Recommendation:** 3

**Summary:**

This paper proposes GTR, a trajectory representation framework built on a pre-train and fine-tune architecture.
The proposed GTR consists of a Multi-View Encoder (MVE) and Spatio-Temporal Fusion Mixture of Experts (ST-MoE), supports pre-training, fine-tuning, and Online Frozen-Hot Updating (OFU), and facilitates multiple downstream tasks.
Compared with previous work, GTR supports POI embedding, dynamic updating, and more downstream tasks. Experiments demonstrate that the performance of GTR surpasses previous work by a large margin, and the proposed modules are effective in most cases.

**Claims And Evidence:**

Yes.

**Essential References Not Discussed:**

No.

**Experimental Designs Or Analyses:**

The experimental designs and analyses are largely thorough and effective.
However, some experimental results did not meet expectations and lack further discussion.

**Methods And Evaluation Criteria:**

Yes.

**Other Comments Or Suggestions:**

Both MLM and Triplet Training are pre-training methods. It is beneficial to list them in parallel in Section 3.2.2 for clarity.

**Other Strengths And Weaknesses:**

Strengths:

1. This paper is highly comprehensive and technically solid. The proposed method supports pre-training, fine-tuning, and dynamic updating; it takes multi-view inputs and facilitates multiple downstream tasks. Experiments demonstrate its superior performance.

2. The proposed challenges are well-justified, and the corresponding solutions are innovative, largely effective, and of further practical value.

3. The paper is well-presented, with a clear flow from the challenges to the corresponding solutions and the purposes of the experimental design.

Weaknesses:

1. Limited details are provided on the online updating approach; please elaborate on it. For example, how does the method process the newly available trajectory data?

2. Some experimental results did not meet expectations, such as those in Table 5 and Table 9. It is necessary to provide further discussions.

3. Multiple downstream tasks and evaluation metrics are presented in the paper, and it is beneficial to provide definitions of the metrics and specify which metrics indicate good performance, for clarity.

**Questions For Authors:**

1. Please see the weaknesses.

2. Would you provide a comparison of model parameters between the proposed GTR and the baselines?

**Relation To Broader Scientific Literature:**

The key contributions of the paper are related to the broader scientific literature on trajectory representation learning and to multiple application studies on downstream tasks, such as travel time estimation.

**Theoretical Claims:**

The application of the related theory in this paper is correct.

---

> ### Author Rebuttal · Authors · 2025-04-01
>
> **We thank the reviewer for offering the valuable feedback. We have addressed each of the concerns as outlined below.**
>
> ```
> W1: Limited details are provided on the online updating approach.
> ```
> We are happy to provide more details about the online updating approach. To process newly arrived trajectory data, we employ the validation set to simulate real-world online/streaming scenarios. The model undergoes single-epoch incremental model updates.
>
>
> ```
> W2: Some experimental results did not meet expectations, such as those in Table 5 and Table 9.
> ```
> We appreciate the reviewer's careful examination of our experimental results. Here, we are happy to provide more detailed experimental discussions to alleviate your concerns.
>
> Regarding **Table 5** (travel time prediction task), our online method GTR\* shows slightly reduced effectiveness compared with our offline method GTR. This can be attributed to the presence of outliers in short-duration trajectories. Due to the sensitivity of regression task to short-term anomalies, it tends to overfit newly arriving anomalous real-time data. Nevertheless, our methods (GTR and GTR\*) perform better than the other comparative baselines.
>
> Regarding **Table 9**'s ablation results for travel time prediction, the complete model shows slightly higher MAPE due to a small number of long-duration outliers. MAPE is known to be more sensitive to errors in lower-value ranges, which might amplify the effect of these outliers.
>
>
> ```
> W3: It is beneficial to provide definitions of the metrics and specify which metrics indicate good performance, for clarity.
> ```
> **These evaluation metrics are standard measures widely adopted in trajectory data processing research.** In the interest of manuscript conciseness, we directly cited the related papers in our manuscript. Here, due to space limitations, we provide rough definitions of the metrics and specify which metrics indicate good performance. We acknowledge that providing more detailed computational formulations could enhance reproducibility, and we will include these in the revised manuscript's appendix to ensure full methodological transparency.
>
> For **trajectory simplification task**, $PED$ computes the shortest perpendicular distance between a deleted point $p_i = (x_i, y_i)$ and the line segment connecting its neighboring points $p_s = (x_s, y_s)$ and $p_t =(x_t, y_t)$. Lower $PED$ indicates lower compression errors.
>
> For **trajectory imputation task**, $Recall@x$ evaluates the model's ability to recover masked tokens by checking whether the ground truth appears in the top-x predicted candidates. Specifically, if the true value is contained within the top-x ranked predictions, $Recall@x$ is assigned 1 for that token; otherwise, it is assigned 0. The final metric is computed by averaging these binary outcomes across all masked tokens in the evaluation set. The MAP represents the probability of the precision, and higher $Recall@x$ and MAP indicate higher performance.
>
> For **travel time estimation task**, we use Mean Absolute Error (**MAE**), Mean Absolute Percentage Error (**MAPE**), and Mean Square Error (**MSE**) metrics. Specifically, lower MAE, MAPE, and MSE indicate higher prediction accuracy.
>
> For **trajectory classification task**, we use $ACC$, $F1-Score$, and $AUC$ metrics. Specifically, higher $ACC$, $F1-Score$, and $AUC$ indicate higher classification accuracy.
>
> For **trajectory generation task**, we use Hausdorff and DTW distance metrics. Lower DTW and Hausdorff value indicate higher performance.
>
> For **trajectory similarity computation task**, we use Mean Rank and $HR@k$ metrics. Lower MR value and a higher $HR@k$ value indicate better performance.
>
>
> ```
> Suggestions: Both MLM and Triplet Training are pre-training methods. It is beneficial to list them in parallel in Section 3.2.2 for clarity.
> ```
> Thanks for the suggestion. Such layout issues can be easily adjusted.
>
>
> ```
> Questions: Would you provide a comparison of model parameters between the proposed GTR and the baselines?
> ```
> A comparison of model parameters is shown below. While GTR has a higher parameter count due to its multi-view encoder (MVE) and spatio-temporal fusion pre-training (STP) modules, this increase is justified by two key advantages. (i) **Enhanced Capability**. The additional parameters enable GTR to support more downstream tasks effectively. (ii) **Performance Gains**. The trade-off in model size is offset by improvements in accuracy and robustness.
>
> |  Model Name |  Parameter Size (MB)  |
> | ---------- | ------------------ |
> | PIM| 94.57|
> | Trembr| 148.08|
> | Toast| 161.18|
> | START| 1126.40|
> | LightPath| 73.96|
> | JGRM| 375.60|
> | GTR| 862.99|

---

> > ### Comment · Reviewer_vfe9 · 2025-04-05
> >
> > Thanks for your response. My concerns have been well-addressed.

---

> > > ### Author Response · Authors · 2025-04-07
> > >
> > > We are happy that our responses have addressed all your concerns. We thank the reviewer for reviewing our paper and providing us with invaluable comments and suggestions!

---

### Decision · Program_Chairs · 2025-05-01

**Decision:**

Accept (poster)

**Comment:**

This paper proposes a trajectory representation framework by integrating free-space trajectories with road network-based trajectories.

After rebuttal, all four reviewers gave positive comments, recognizing the good performance and novel idea of this paper.

The Area Chair (AC) agrees with the reviewers and recommends accepting the paper. Additionally, it is strongly suggested that the authors incorporate the details discussed in the feedback to make the paper more comprehensive.